# Optimization of power management in PV-based smart grids using grid-support and grid-forming inverters

Amam Hossain Bagdadee[1,2*], Ishtiak Al Mamoon[3], Deshinta Arrova Dewi[4], A. K. M. Muzahidul Islam[5], Li Zhang[1]

**1** School of Electrical and Power Engineering, Hohai University, Nanjing, China, **2** Department of Electrical and Electronic Engineering, Presidency University, Dhaka, Bangladesh, **3** Department of Computer Science and Engineering, International University of Business Agriculture and Technology, Dhaka, Bangladesh, **4** Department of Computer Science and Engineering, United International University, Dhaka, Bangladesh, **5** Faculty of Data Science and Information Technology, INTI International University, Nilai, Malaysia

* a.bagdadee@hhu.edu.cn

## Abstract

This paper addresses the synthesis and analysis of advanced control strategies in photovoltaic (PV) based smart grids with distributed generation, focusing on grid support and grid-forming inverters. The penetration level of renewable energy sources such as photovoltaic (PV) systems is on the rise, and maintaining grid reliability, smooth power management, and secure power-sharing has become difficult. It is anticipated that the control approaches are able to optimally allocate power among PV generation, energy storage systems and load without compromising on grid-tied stability in both on-grid and island modes of operation. These kinds of grid-support inverters can offer essential ancillary services as voltage and frequency regulation, which should be cost-effectively supported in a grid-connected operation. In isolated mode, however, the grid-forming inverters are in normal operation. The novel optimal power-sharing strategy developed in this paper can further achieve real-time coordination of distribution generation units' power allocation, to reduce dependence on PV output variation. Results from the simulations, carried out using Simulink, demonstrate that these methods result in direct assistance in grid operation, loss reduction and system restoration. Based on the results, the proposed system has a superior ability in terms of power regulation, stability and quality control than conventional methods.

## 1. Introduction

The integration of photovoltaic (PV) systems in power networks leads to an enormous change in the energy landscape around the world. Solar is one of the most abundant and available forms of renewable energy. PV deployment in residential and

**Data availability statement:** The datasets generated and analyzed during this study are not publicly available due to ethical and institutional restrictions. Although de-identified, the data contain infrastructure-sensitive operational and system configuration information that cannot be openly shared under the data governance and research ethics policies of the host institution. These restrictions were imposed by the Institutional Research Ethics Committee of Presidency University, Bangladesh. Data access requests may be submitted to the Office of Research and Innovation, Presidency University, Bangladesh (email: office.research@presidency.edu.bd). The authors are not permitted to share the data directly.

**Funding:** This study was partially supported by the Institute for Advanced Research (IAR), United International University (UIU), Bangladesh, in the form of a research grant awarded to AKMMI (Grant No. IAR-2025-Pub-105). The funders had no role in study design, data collection and analysis, and preparation of the manuscript. No additional external funding was received for this study.

**Competing interests:** The authors declare that they have no competing interests.

utility installations has grown substantially. The primary reason behind this increase is the increasing requirement to reduce GHG emissions, achieve renewable energy targets, and provide a low-carbon alternative to hydrocarbon fuels [1]. As nations aim to decarbonize their power systems, PV is a cornerstone of smart grids – the future grid with enhanced contribution from renewable and digital sources for more efficient electricity generation on a distributed basis—technical challenges for integration of PV systems to smart grid [2]. One of the main hurdles is that solar energy can be variable and intermittent. Solar energy generation inherently depends on solar power, which fluctuates due to differences between day and night and seasonal changes, while indecisive random weather events remain [3]. These variations can lead to grid instability, which could cause voltage and frequency regulation problems in the power system. PV systems have achieved high penetration levels in some areas, resulting in over-generation during peak solar output and under-generation when the solar resource is low, increasing complications in power management. Power flow management becomes increasingly complex, particularly when power generation is distributed at many small-scale decentralized locations (residential rooftops or community solar farms), which no longer connect to any single central hub [4,5]. Centralized power generation and unidirectional flow of electricity were the architecture on which traditional power grids were designed to operate. Nowadays, with the integration of PV systems and other renewable energy sources, these must operate on bidirectional power flow. These advanced control strategies manage real-time generation and consumption to balance dynamically with grid stability, thereby avoiding power quality problems [6]. Inverter technology is paramount when converting the direct current (DC) output from PV panels into alternating current (AC) that can be used by a grid or consumed locally.

The grid-forming and grid-support inverters are two of the most essential inverter types for PV-based smart grids. Grid-support inverters can supply various ancillary services to help stabilize the grid, such as voltage regulation, reactive power compensation, and frequency response [7,8]. It is linked to the primary grid and allows better utilization of weak renewable energy sources. The grid-forming inverter also generates and maintains the voltage provided to an off-grid or islanded situation when not connected to the main utility [9,10]. PV-powered microgrid operates standalone with the help of grid-forming inverters that generate a reference signal for other inverters and loads to ensure a stable energy supply without relying on the primary grid. The PV systems have increasingly important roles in power grids, and due to these technical challenges related to their integration with the grid, there is an urgent necessity for advanced control methods that can perform effective coordination of generation-side and distribution-side management, and storage at demand levels properly. These strategies should allow efficient power sharing among distributed generation units, provide grid support when solar outputs fluctuate, and maintain the overall performance of the power system. The increasing integration of photovoltaic (PV) systems into the smart grid introduces significant uncertainty and variability in the produced power, which is a direct function of the stochastic nature of weather conditions and solar radiation. This diversity makes the grid stabilization, continuous

voltage supply and perfect power management planning also more complex. The classic deterministic optimization approach does not actually take this uncertainty into account and may lead to non-optimal or unfeasible operational decisions when the probabilistic implementation of the process is considered [11]. From these concerns, research has been exploring more powerful higher-order models to model uncertainty, providing robust and adaptive solutions. Stochastic programming is one of the most utilized approaches, where several realizations of the uncertain PV output are characterized by a random variable, and optimization problems for minimizing the expected cost or maximizing the reliability can be formulated. Within scenario-based methods, the framework can be expanded so that a finite number of representative scenarios, often constructed from historical observations or forecast models, exist in such a way as to accommodate the diversity of PV output fluctuations. Such scenarios are added to the optimization in order that system operators would be able to simulate strategies for a variety of conditions, but without diverging into computational complexity. The risk-sensitive strategies, such as Conditional Value at Risk (CVaR), are proposed, which take explicitly into account rare but relevant extreme events like acceptably long periods without solar generation and large drops in the available solar power [12]. The optimization processes are based on the CVaR risk measure, and they propose diminishing the adverse consequences of worst scenarios by penalizing an excessive level of risk taking, hence contributing to system resiliency and securing. With these uncertainty models in the development of smart grid power management, a complete decision-making framework is formulated, which presents the considered control strategies as mainstream ones within high-penetrated renewable-based energy systems.

## 1.2. Research objectives

This research aims to develop power management and distributed generation control strategies in PV-based smart grids. Integrating grid-support and grid-forming inverters with sophisticated power electronics, addressing the technical issues related to higher PV penetration with increased performance of inverter-based generation, focusing on ensuring grid stability and power quality, this paper provides research innovation. The research has the following specific objectives:

- Developing real-time control strategies that maximize the utilization of PV generation, energy storage, and grid resources to enable efficient power management in both grid-connected mode and islanded operation.

- Develop advanced control algorithms that enable grid-supporting inverters to provide beneficial ancillary services such as reactive power support and frequency regulation in a fast-responding fashion with variability in solar output.

- Create new power-sharing methodologies between distributed generations and energy storage systems, which would implement the distribution of electricity on the grid in a smart way, nearly optimally, without violating fairness constraints under time-varying PV output levels caused by natural environment situations.

- Novel control strategies for grid-forming inverters that strengthen resilience in islanded mode by developing management methods of microgrids capable of operating independently from the primary grid and with stable voltage and frequency

- System efficiency maximization involves designing optimization algorithms that minimize energy losses and operating costs, and enhance the overall energy efficiency of the PV-based smart grid.

## 1.3. Scope of the study

The objective of the proposed research work is to investigate control strategies for power management in smart grids supplied from photovoltaic (PV) based distributed generation units. The importance of grid-forming and grid-support in effectively operating a secure distribution smart grid, which can adapt load demands and PV output, changing from island and grid-connected modes, is emphasized by this study. The major features addressed in this study are:

- This research presents the evaluation and management of the integration of PV systems in distributed generation networks, including issues like solar power intermittency and variability.

- The study will investigate the capabilities and control strategies of grid-support inverters and grid-forming inverters in maintaining grid stability, supporting grid operations, and enabling the islanding operation of microgrids. This study will investigate how these inverters can be further enhanced to improve overall system performance.

- Advanced control algorithms will be developed to govern the dynamic response of PV generation, energy storage systems, and power consumption in the smart grid environment. It will have real-time power management, voltage and frequency Regulation, and power-sharing mechanisms.

This research will focus on developing sustainable energy systems that support even higher proportions of PV integration and ensure grid stability, efficiency, and reliability. The results of this study would offer valuable information for smart grid planning and the widespread adoption of renewable energy technologies.

## 2. Literature review

The deployment of solar Photovoltaic (PV) technology has expanded at an unprecedented pace over the past 30 years, and this rapid growth is primarily attributed to improvements in cost reductions, cell efficiencies, and government incentives [13,14]. Distributed generation (DG) represents a cogwheel menu in this shift from centralized power systems, with large-scale conventional gigawatt power plants being responsible for almost all electricity output, to decentralized systems where small-scale changes to renewable-based energy sources are contributing even a part of the total power system. PV-based distributed generation is critical to this shift since solar energy is available, clean, and cost-competitive with fossils [15]. Solar panels had historically been installed at off-grid locations, whose primary energy requirement was to be provided by the electricity produced via a grid [16]. However, due to the significant incentives to reduce greenhouse gas emissions and promote independence from fossil fuels, PV systems are increasingly becoming competitive or preferred for grid-connected distributed generation. PV systems are increasingly being coupled with other technologies, such as advanced monitoring and communication infrastructure, to enable the integration of these installations into smart grids [17]. PV systems are widely deployed in residential, commercial, and utility-scale applications, contributing significantly to the global energy mix. The integration of PV-based distributed generation offers several benefits. PV generation is clean and renewable, which can save fossil energy and reduce carbon emissions. Distributed PV systems make energy more resilient by decentralizing its production and, hence, are less vulnerable to large-scale blackouts at the power-system level. As the price of solar panels and other technologies drops, large-scale PV systems offer substantial cost savings to consumers while saving utilities money. However, several challenges complicate the widespread deployment of PV-based distributed generation. Solar power is intermittent because it relies on sunlight, which varies daily and seasonally. This variability poses several complications to grid stability, especially during high PV penetration. High penetration of PV systems can cause voltage fluctuations and harmonic distortion, thus affecting the power supply to the consumer. The bulk of the grid was not initially designed for distributed generation or bidirectional power flow. Extensive modernization, mainly by the control and communication system operation of distributed PV generation, is required with the current infrastructure.

### 2.1. Smart grids and power management

Smart grids combine digital technologies, communication systems, and automation that enable the integration of generation, distribution, and consumption to facilitate overall efficient management of electricity operations [18,19]. The traditional power grid depends on centralized generation, which allows a one-way flow of electricity. In contrast, smart grids are more suitable for renewable energy sources like PV systems with bidirectional power flow. Smart meters that provide real-time data on energy consumption and generation. Energy Management Systems (EMS) that monitor and control energy production, distribution, and storage. Technologies that enable dynamic energy use in response to grid conditions. The

smart grid will connect more renewable energy resources, reducing variation and intermittency in source operation, such as Solar or wind, using the latest control systems [20]. Power management in the smart grid must balance generation with demand in real-time, ensure voltage and frequency stability, and optimally utilize distributed energy resources [21]. Smart grids also enable demand-side management through consumer response to price signals from the grid, reducing or eliminating expensive peaking power plants and improving overall grid efficiency.

## 2.2. Grid-support inverters

Grid-support inverters are essential for the stability of smart grids due to the high penetration levels of PV-based distributed generation [22]. Inverters that support the grid achieve this more extensive grid-support functionality to ensure power system stability while converting PV systems' direct current (DC) output into alternating, grid-compatible power quality. Grid-support inverters can control the reactive power inserted into the grid, providing support voltage for increased reliability levels that can be upset by fluctuating PV generation [23]. Grid-support inverters can change their active power output to help keep the grid frequency stable, as imbalances between generation and consumption affect the regional supply situation [24]. Grid-support inverters can supply or absorb reactive power, which provides voltage control and improves the overall quality of electricity [25]. Several studies have researched the control strategies and operation of grid-support inverters in different applications to enable efficient use with smart grids requiring distributed PV systems [26]. Several methods have been studied to enhance grid-support characteristics of the renewable-based inverter systems. The new inverters are also providing new functions that maintain the inverter online for short-duration faults or transient faults through voltage ride-through functions, therefore enhancing power grid protection. Furthermore, the literature studied better algorithms for providing stable dynamic frequency control in microgrids, especially when PV production is variable [27,28]. This allows for the control of virtual synchronous generators (VSG), designed to enable inverters to mimic traditional behavior from a synchronous generator and thereby make the inverter vastly more capable of providing stable power during system disturbances. [29] Other recent work has been done, targeting VSG, with the goal of being able to help the inverter contribute much larger quantities of stable power when there are disturbances on the system. These properties have made a grid-friendly inverter a necessary part of any future PV-based smart grid. The issue of stability and security operation for power systems with a high penetration of inverter generation retracts interest towards the Virtual Synchronous Generator (VSG) or simply virtual inertia. Recent comprehensive studies and field experiences demonstrate that VSG control approaches—simulating synchronous-machine behaviors with tunable virtual inertia and damping—in larger transient perspectives offer better frequency performance to the systems penetrated by high PV contents, but also involve trade-offs in circulating currents, control complexity, and parameter tuning, which are scenario dependent.

## 2.3. Grid-forming inverters

Grid-forming inverters function independently to set up and uphold the grid voltage and frequency, which is not valid for grid-support inverters [30,31]. The grid-forming inverters issue a reference signal when not connected to the primary grid in islanded or microgrid settings. It is critical to operate reliably when the main grids are unavailable or in case of blackouts [32]. Microgrids and distributed generation require grid-forming inverters that can operate autonomously. Integrated with islanded microgrids that can be deployed on time to reduce hourly emissions and charges [33,34]. The inverters form a virtually coordinated, reliable voltage and frequency reference against which other items in the microgrid can synchronize and work stably [35,36]. Maintaining voltage and frequency constant in islanded mode is challenging, especially if there is an intermittent PV generation penetration [37]. High-level coordination with grid-support inverters, including grid-forming and grid-support-type inverters, must be carefully managed to avoid disruption during the transition between grid-connected and islanded modes [38,39]. Grid-forming inverters must meet the needs for both load and generation, which could raise costs for small microgrids.

## 2.4. Existing control strategies for power management in smart grids

The literature is abundant in AI-driven and learning-based control techniques for inverter and microgrid networks management. Further survey and application papers from say 2023–2025 show that machine-learning approaches including supervised learning for predictive control, reinforcement learning for adaptive set pointing, and hybrid model-based versus data-driven controllers can lead to more advanced decision-making in real time that can handle changing conditions better while relying less on conservative regulation that can bring problems with safety securities, interpretability of solutions, and need for rigorous training and validation using edge-case scenarios [12,40].

### 2.4.1. Conventional control strategies.
Power management has been a traditional control strategy, balancing generation and demand to maintain voltage and frequency in real-time. Centralized control was traditionally prevalent in most grids, where some central entity controlled power flows and dispatched individual units of generation [41,42]. This approach is no longer suitable in the modern smart grid, especially with distributed generation, because renewable sources require more flexible and dynamic control strategies.

### 2.4.2. Modern control strategies for power sharing and load management.
Modern smart grids with distributed generation require decentralized control strategies, allowing real-time adjustments based on local conditions. There are some modern control strategies. Droop control is a standard method in microgrids to determine the output power of inverters based on measured local frequency and voltage so that multiple inverters supply power proportionally [43,44]. Model predictive control is an optimization-based control strategy considering future load and power flow generation variations. Decentralized and hierarchical control is a type of strategy that achieves distributed individualized goals as each inverter or generation unit operates on local measurements, minimizing the requirements for central control [45]. With hierarchical control, a combined coordination layer ensures that local actions align with global system objectives. These control strategies are essential for managing the variability of PV generation and ensuring stable operation of smart grids, particularly as the penetration of distributed energy resources grows.

Finally, distributed energy storage allocation and risk-aware optimization have become central topics for integrating PV at scale. Recent works also recommend systematic approaches for placement, size and schedule of distributed storage based on evolutionary and mathematical programming formulations, with clear advantages to reduce curtailment and enhance local voltage and frequency support. Risk-aware formulations using Conditional Value-at-Risk (CVaR) and two-stage stochastic programming are becoming mainstream for modeling low-probability, high-impact events such as extended low irradiance and storm-induced outages, and to make economically robust storage y and operating decisions. These threads of work on VSGs, AI control and CVaR-informed storage optimization suggest promising hybrid concepts but also highlight a gap in the literature: scalable integrated frameworks that incorporate VSG and grid-forming control, provably safe AI-adaptive strategies, and risk-type constraints in such systems remain limited.

## 3. System architecture

Photovoltaic (PV) and energy storage systems are a few of the many distributed generation units that a smart grid can have in conjunction with various other units. It operates with an advanced energy management system (EMS) (Fig 1). The grid needs to be flexible enough to cope with dynamic power flows while maintaining stable voltage and frequency and balancing generation against consumption efficiently. Photovoltaic (PV) Systems are one of the distributed energy resources that utilize sunlight to generate electricity. Inverter devices convert the solar panels from DC to alternating current, as AC connects PV/ Solar systems with grids. Despite the intermittent power generation faults caused by solar irradiance fluctuations, PV systems are increasingly used. Energy storage — particularly batteries for the time being- is instrumental in managing solar intermittency by storing surplus generation at peak hours and discharging it during low generation or peak demand. The energy management system is to independently control the energy stored in energy storage systems to eliminate power fluctuations and improve grid stability. Inverters interface between PV systems, energy storage systems, and the grid. It converts DC to AC, which allows for power communication and control. Grid-forming

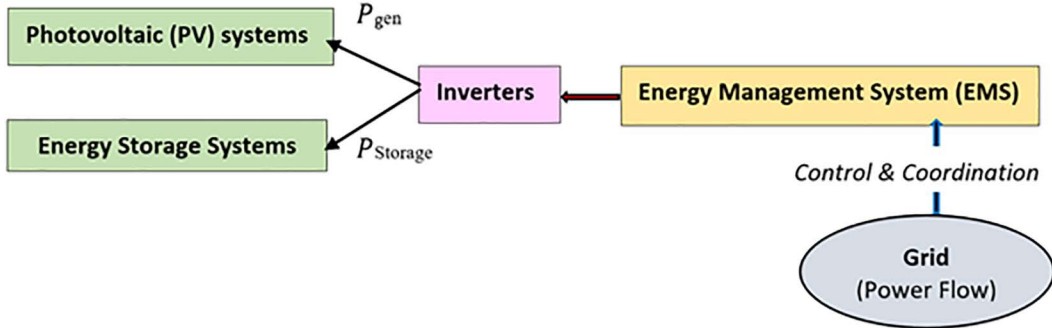

**Fig 1. Smart Grid Architecture.**

and grid-support inverters are the two most common inverters that perform specific functions to maintain grid stability (Fig 2). The energy management system coordinates these elements in real-time, managing power flows, generation, and consumption. It ensures the stability of a grid in terms of power generation, voltage regulation, and load sharing using a sophisticated control algorithm. Mathematically, the power balance in a smart grid can be expressed as:

$$P_{gen} + P_{storage} = P_{load} + P_{loss} \tag{1}$$

Equation (1) describers the power generated $P_{gen}$ by photovoltaic (PV) systems and other distributed energy resources; $P_{storage}$, which refers to the power supplied or absorbed by energy storage systems; $P_{load}$, representing the power demanded by loads connected to the grid; and $P_{loss}$, accounting for transmission and conversion losses. These factors demonstrate that the energy produced and saved equals the energy used by the load and losses in this aspect, which is a crucial factor for the stability of smart grid systems.

### 3.1. Grid-support inverter configuration

Grid-support inverters work in grid-connected mode to offer ancillary services such as voltage, frequency, and reactive power control. These inverters do not form the grid but help stabilize it by adjusting their output in response to grid

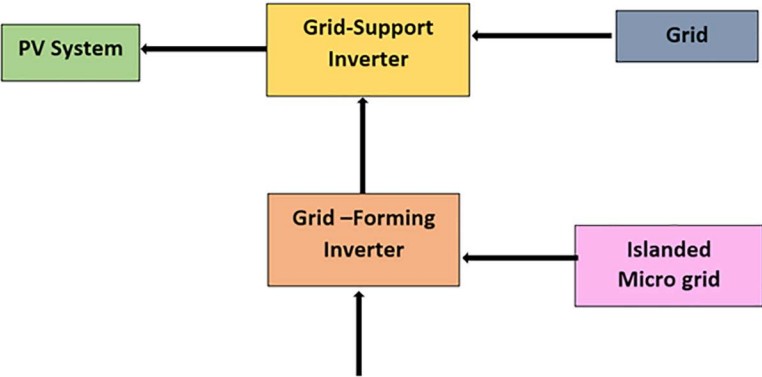

**Fig 2. Grid-Support Inverter and Grid-Forming Inverter.**

conditions. Grid-support inverter controllers generally require real-time monitoring of variables such as the voltage and frequency modulated into the grid, utilizing control strategies. The grid-support inverters, the power output can be:

$$P_{gs}(t) = P_{pv}(t) - P_{reactive}(t) \qquad (2)$$

Equation (2) describes $P_{gs}$ (t), which represents the active power transferred by the grid-support inverter at time t. It is given as the net difference between power delivered by PV $P_{pv}$ (t), offset from reactive power compensation $P_{reactive}$(t) used for voltage support. In this equation, $P_{pv}$ (t) is the instantaneous photovoltaic power output. By contrast, $P_{reactive}$ (t) is the de-rated inverter capacity set aside to reinforce voltage stability by reactive power control. Integrating an added energy and ancillary service model ensures that the grid-supporting inverter not only injects active power but also provides system support services in voltage regulation, further solving a two-component optimization problem; thus brings both supply of energy and provision of services into balance within the smart-grid system.

The voltage regulation by grid-support inverters is governed by the droop control law, where the inverter adjusts its reactive power output $Q$ in response to voltage deviations $\Delta V$:

$$Q = Kv \cdot \Delta V \qquad (3)$$

Equation (3) governs how the grid-support inverter adjusts its reactive power $Q$ based on voltage deviations ΔV. The voltage droop coefficient $Kv$ determines the sensitivity of the inverter's response to changes in voltage. When voltage deviates from its nominal value, the inverter adjusts$Q$ to restore stability, helping to regulate and maintain voltage levels in the system.

### 3.2. Grid-forming inverter configuration

Grid-forming inverters are used in Islanded mode, with no primary grid to establish and maintain each voltage & frequency profile. Autonomously, these inverters imply a reference signal to other generation units and loads that need to synchronize with the source. Grid-forming inverters are essential for microgrids and black start power restoration, which must be achieved without relying on power from external sources. The control strategy for grid-forming inverters is a voltage source control, in which the inverter controls the voltage and frequency at its output. The grid-forming inverter's active and reactive power equations are typically written as:

$$Pgf(t) = V(t) \cdot I(t) \cdot cos\varphi(t)$$
$$Qgf(t) = V(t) \cdot I(t) \cdot sin\varphi(t) \qquad (4)$$

The active and reactive power generated by the grid-forming inverter based on terminal voltage, current, and power factor angle is given by Eqn. (4). The active power is the real part of the inverter output that directly contributes to meeting loads and ensuring power system frequency stability. The reactive power, on the other hand, is the imaginary part, and it supports voltage relay and reactive compensation within the grid. Where V(t) is the instantaneous RMS voltage of the inverter terminal, I(t) is the instantaneous RMS current sourcing from the inverter, and φ(t) is the angle that voltage leads or lags current that the phase difference between the source voltage and source current, typically used to represent the power factor. Combined, these representations guarantee that the inverter supplies both real and reactive power components according to what is required by the grid, allowing it to stabilize under load and generation conditions.

$$fgf(t) = fref + Kf \cdot \Delta Pload \qquad (5)$$

The frequency $Fgf(t)$ output of the grid-forming inverter is governed by equation (5), varying to accommodate a change in load demand $\Delta Pload$. The frequency is adjusted for power imbalances based on droop control in the inverter, whereas $Kf$

is a coefficient responsible for frequency drop and controls-related tolerances. Grid-forming inverters are required for long-term voltage and frequency stability to operate microgrids reliably in an autonomous mode.

## 3.3. Energy management system and control logic

The energy management system is the supervising controller of PV generation, the energy storage unit, and the inverter to promote the reliable and economic operation of a smart grid. High-level control for it is of a hierarchical and modular structure. The energy management system is continuously updating the prediction of PV generation along with load demand and grid state in real time by measurement and communication signals. The energy management system decides on the power balance by dispatching how much of the load is served from PV, storage discharging and grid import and export. When the PV generation is greater than the demand, the energy management system concentrates on local consumption, stores surplus energy in the energy storage systems and cuts off excessive power as energy storage becomes saturated. It discharges its stored energy before importing from the grid and optimizing for both economic cost and system stability. The energy management system makes use of a grid-support and grid-forming inverter with inverter coordination. During islanded operation, voltage and frequency references are established by grid-forming inverters using droop control. Grid-support inverters, on the other hand, regulate their active and reactive power using a control mechanism to counteract voltage and frequency deviations from set points. The EMS maintains the constraints similar to real-time constraints, such as SOC limits, inverter power ratings and grid security thresholds for voltage and frequency. It also utilizes adaptive logic in response to variations in communication delays and noise level, and its stability is ensured by modifying the set points of inverters as the magnitudes of delayed measurements when necessary. The energy management system is a real-time optimization and coordination machine which negotiates economic goals, minimizing energy cost and deterioration with technical imperatives: voltage level, frequency control and security of the network. This hierarchical control strategy allows the system to operate in stable conditions and accept PV intermittency and load-demand uncertainty under grid-connected and islanded operation.

## 3.4. Problem formulation

The power management issue in a PV-based smart grid is set as the objective of optimization for optimized power sharing, voltage regulation, and system stability with low energy losses while securing reliable operation of both grid-support and grid-forming inverters.

### 3.4.1. Objective function.
The optimization problem is formulated as the minimization of total system cost, including power loss, energy storage, and costs associated with voltage deviation. The optimization problem is set up to minimize the total system cost, including power loss, energy storage, and voltage deviation costs, which can be expressed as:

$$Minimize\ J = \sum_{t=1}^{T} [Closs\,(Ploss\,(t)) + Cstorage\,(Pstorage\,(t)) + Cvoltage\,(\Delta V\,(t))]$$

(6)

Equation (6) represents the formulation of the total system cost at time t, which is a sum of three main factors to be minimized by the optimization problem. The first term, Closs(Ploss(t)), models the power loss in the network with respect to active power loss Ploss(t), which has a direct effect on the efficiency of the system. The second term, Cstorage (Pstorage (t)), models the cost of charging and discharging the energy storage system; here, Pstorage (t) denotes the battery power at time t corresponding to round-trip efficiency and cycling costs. The third term, Cvoltage (ΔV(t)), considers the penalty of voltage deviation, where ΔV(t) is the distance to nominal reference limits for secure operation in terms of voltages. In combination, these variables define a cost-minimization problem over time horizon T, whereby the optimal solution will seek to balance PQ, reliability and economically.

### Constraints

- **Power balance constraint**: The total power produced must equal the total load. This ensures a stable operation of the system, avoiding frequency deviations. The total of all power sources (solar, wind, battery, etc.) must mathematically equal the sum of all loads(consumption) combined.

$$Pgen(t) + Pstorage(t) = Pload(t) + Ploss(t) \tag{7}$$

Equation (7) ensures that the total power generated by the PV system and energy storage at any time $t$ equals the power demanded by the load and the losses in the system. It maintains the system's power balance for reliable operation.

- **Inverter power limits**: Inverter power management capability is a limiting factor. Power feeding into the grid or loads should not exceed the rated limits; otherwise, the inverter hardware will overheat and break down.

$$Pgs(t), Pgf(t) \leq Pinverter, max \tag{8}$$

This Equation (8) ensures that the power output from both the grid-support $Pgs(t)$ and grid-forming inverters $Pgf(t)$ does not exceed the maximum rated power of the inverter, $inverter, max$. This protects the inverters from being overloaded.

- **Voltage regulation constraint**: The voltage levels at different nodes in the system must be kept within specified bounds (typically ±5 percent of the nominal voltage). It ensures the demand for electricity for equipment and reduces the potential of voltage instability.

$$Vmin \leq V(t) \leq Vmax \tag{9}$$

This Equation (9) ensures that the system voltage $V(t)$ stays within the specified operational limits, $Vmin$ and $max$, which is essential for maintaining grid stability and avoiding voltage-related issues in the system.

- **Energy storage dynamics**: Battery energy storage systems are limited by the state-of-charge, charge and discharge rate, and the efficiency losses. The dynamics of the battery system must be preserved by control strategies to protect the battery's health and ensure energy when energy demand is high or when there is a shortage of power available.

$$Estorage(t+1) = Estorage(t) + \eta charge Pstorage(t) \cdot \Delta t - \frac{Pstorage(t) \cdot \Delta t}{\eta discharge} \tag{10}$$

Equation (10) models the behavior of the energy storage system, where the stored energy $Estorage(t)$ is updated based on the power stored or discharged over time $\Delta t$. The charge and discharge efficiencies $\eta charge$ and $\eta discharge$ account for losses during energy transfer. This optimization problem can solve the balance of power generation, storage utilization, and load demand to ensure grid stability while achieving efficient electricity consumption.

## 4. Control strategy development

In PV-based smart grids, powerful control strategies can be formulated to manage suitable power flow, system grid balance, and the behavior of distributed energy resources such as PV systems, inverters, and energy storage, which are critical. This section elaborates on the control algorithms for real-time power management, emphasizing grid-support and grid-forming inverter controls and load-balancing coordination frameworks to perform energy distribution across the system.

## 4.1. Power management control strategies

Dynamic power management in PV-enabled smart grids aims to balance energy generation, consumption, and storage with minimum system losses while maintaining grid stability (Fig 3). This involves real-time control algorithm improvements for power output to PV systems, energy storage unit management, and low disturbance operation under load requirements.

### 4.1.1. Energy balance equation.

At any time $t$, the total power generated by the PV system and energy storage must equal the load demand plus losses:

$$Ppv(t) + Pstorage(t) = Pload(t) + Ploss(t) \tag{11}$$

Equation (11) ensures that at any given time $t$, the total power generated by the PV system $Ppv$ and the energy storage system $Pstorage(t)$ matches the total power consumed by the loads $Pload(t)$ and the power lost $Ploss(t)$ due to inefficiencies. This sets up an equation that ensures the grid functions safely and efficiently, delivering stable power to flexible consumer demand while minimizing losses.

### 4.1.2. Objective function for power optimization.

The primary objective is to minimize system losses and optimize energy consumption. The objective function for real-time control can be defined as:

$$min\,(Pstorage, Ppv)J = \sum_{t=1}^{T}(Ploss(t) + Cstorage(Pstorage(t)) + Cunmet(Pload(t))) \tag{12}$$

Equation (12) indicates the term $Ploss(t)$ transmission and conversion losses during power transport. On the contrary, $Cstorage(Pstorage(t))$ characterizes all the expenses of energy storage systems operation by including both charging and discharging. In additional scenarios, $Cunmet(Pload(t))$ discourages unmet demand in the system as energy generated and stored is just not enough to meet load demands. These components are described to include the goal of energy storage and PV generation optimization, minimizing losses, and maximizing efficient grid operation.

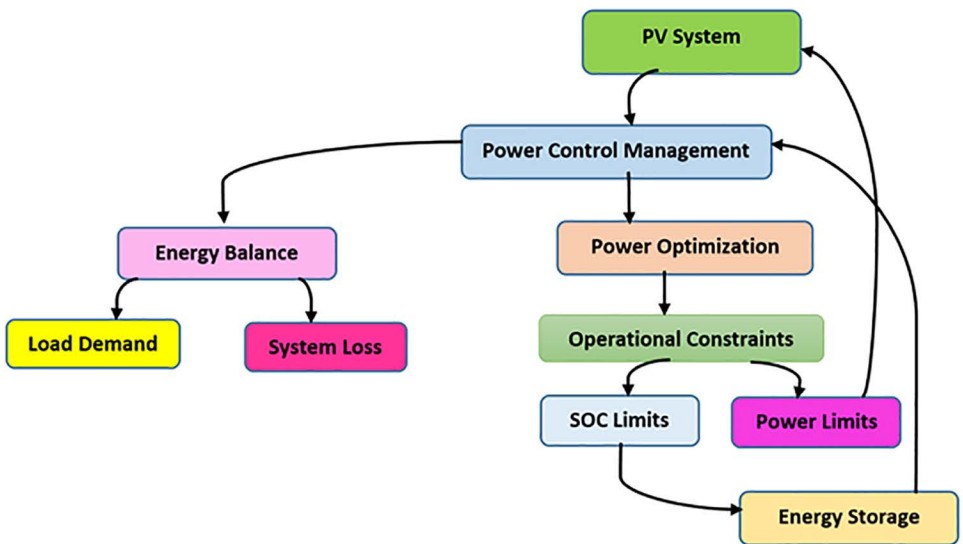

**Fig 3. Power Management Control Strategies in a PV-based smart grid.**

In the power management control strategies of a PV-based smart grid, a conceptual diagram can effectively illustrate the structural energy balance and its interactions, incorporating an objective function with constraints on these components. Fig 3 outlines the energy flow and several other control strategies more suitable for power generation, storage, and demand, while accounting for losses and are limited to system regulations. The following diagram provides a conceptual understanding of balancing and optimizing power in those systems.

## 4.2. Control strategies for grid-support inverters

Grid-support inverters are essential to provide grid stability, especially when the output of PV fluctuates because irradiation varies. These inverters can offer voltage support, frequency regulation, and reactive power capability.

### 4.2.1. Voltage and frequency regulation.
Grid-support inverters adjust their output dynamically in grid-connected mode to help maintain the stability of voltage and frequency support for the system. The control strategy uses droop control, in which an inverter regulates power output in response to voltage and frequency set point variation.

$$Pgs\,(t) = Pref - Kp\,(\omega grid\,(t) - \omega ref) \tag{13}$$

Equation (13) controls the active power $Pgs\,(t)$ provided by the grid-support inverter according to the deviation from reference frequency $\omega grid\,(t)$. The droop coefficient $Kp$, identifies the variation of active power for frequency deviations and helps in grid frequency stabilization.

$$Qgs\,(t) = Qref - Kq\,(Vgrid\,(t) - Vref) \tag{14}$$

Equation (14) corresponds to the reactive power $Qgs\,(t)$ output by the inverter and varies according to the difference voltage between the measured grid voltage $Vgrid\,(t)$ and the reference voltage $Vref$. The droop coefficient $Kq$ factor is related to the sensitivity of reactive power adjustment, which directly connects with voltage regulation.

### 4.2.2. Reactive power compensation.
Grid-support inverters also provide reactive power compensation functionality. The inverter provides the following dynamic control of reactive power injection to support grid voltage:

$$Qgs\,(t) = Kv\,(Vgrid\,(t) - Vref) \tag{15}$$

Compensation of reactive power $Qgs\,(t)$ by the grid-support inverter is controlled as per Equation (15), which dynamically injects or absorbs a desired amount based on voltage deviations $Vgrid\,(t)$ from the reference value $Vref$. The voltage droop coefficient $Kv$ regulates the operation of the control signal to stabilize grid voltage. These control strategies ensure voltage and frequency regulation, enhancing grid stability in standard and high renewable energy integration scenarios.

Fig 4 shows the control strategies for grid-support inverters. The top plot shows the solid blue curve representing how active power $Psg\,(t)$) adjusts based on grid frequency deviations, following the droop control law. The dashed red curve shows how the reactive power $Qgs\,(t)$ adjusts based on grid voltage deviations governed by the voltage droop control (*Top Plot*). The bottom plot shows the green curve demonstrating that reactive power compensation $Qcomp\,(t)$ dynamically adjusts reactive power to stabilize grid voltage as it fluctuates. This compensatory behaviour is essential for maintaining voltage stability, especially in distribution networks with high levels of $PV$ penetration. This visualization highlights how grid-support inverters contribute to grid stability by regulating and compensating active and reactive power.

## 4.3. Control strategies for grid-forming inverters

Grid-forming inverters are required to provide secure voltage and frequency for the islanded grid or off-grid solutions, which include microgrids. Synchronous inverters control and regulate the grid for frequency and voltage reference so that other generation units or loads can synchronize accordingly.

### 4.3.1. Voltage and frequency regulation in islanded mode.

Voltage and frequency are regulated autonomously in islanded microgrids by grid-forming inverters. The control strategy is built by combining the voltage source control and frequency droop-based approach:

$$Vgf(t) = Vref - Kv \cdot (Pgf(t) - Pref) \tag{16}$$

Equation (16) shows that the output voltage $Vgf(t)$ decreases if the power output $Pgf(t)$ exceeds the reference power $Pref$. The coefficient $Kv$ determines how sensitive the voltage is to changes in power output.

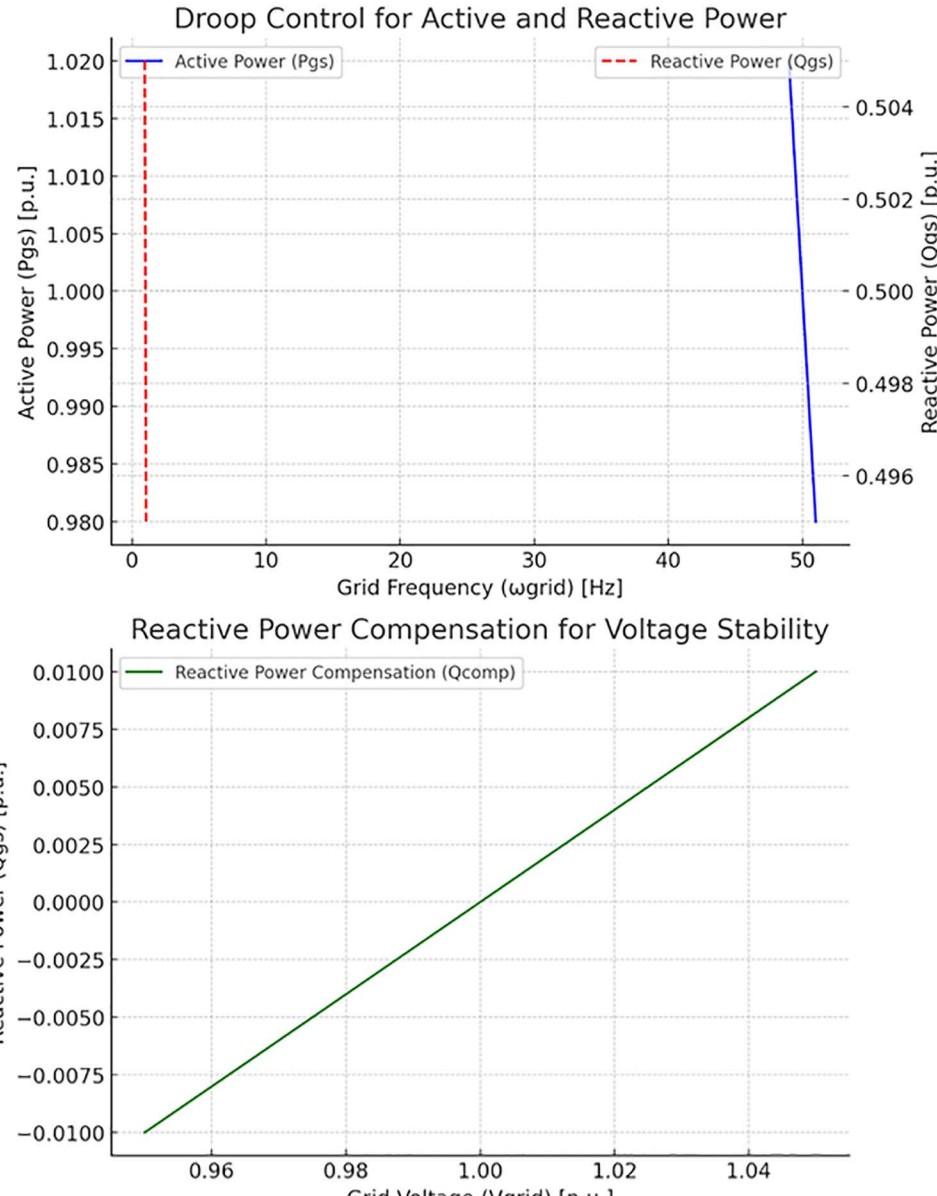

**Fig 4. Control strategies for grid-support inverters.**

$$fgf(t) = fref - Kf \cdot (Pgf(t) - Pref) \tag{17}$$

Equation (17) shows that the output frequency $gf(t)$ decreases if the power output $Pgf(t)$ is higher than the reference power $ref$. The droop coefficient $Kf$ controls the sensitivity of the frequency to power changes. Equations (20) and (21) ensure that the inverter can automatically balance the power to adjust voltage and frequency, stabilizing the microgrid's operation without centralized control. Voltage source control and frequency droop control make it an integrated and autonomous way to adjust voltage and frequency depending on the power generated.

**4.3.2. Virtual inertia for grid stability.** Grid-forming inverters can simulate the natural inertia of synchronous generators to stabilize islanded systems further. This strategy temporarily absorbs or injects power based on the frequency oscillations in response to the rate of change at frequency:

$$Pinertia(t) = -M\frac{d}{dt}\omega(t) \tag{18}$$

Equation (18) involves vital components related to the dynamics of power systems. Here, $M$ is the inertia constant, which reflects the system's resistance to changes in frequency. The term $\frac{d}{dt}\omega(t)$ represents the rate of change of the system frequency over time. $Pinertia(t)$ denotes the power that is temporarily absorbed or injected by the inverter to maintain stability, compensating for fluctuations in the system frequency.

Fig 5 demonstrates the voltage and frequency regulation in islanded mode. The top plot shows the solid blue curve representing how the inverter's output voltage $Vgf(t)$ changes as a function of power deviation from the reference $Pref$ based on the voltage droop control. The dashed red curve shows how the inverter's output frequency $fgt(t)$ changes with power deviation, following the frequency droop control. The bottom plot shows the solid green curve that represents the rate of change of frequency $\frac{d\omega}{dt}(t)$, which oscillates over time. The dashed magenta curve shows the power response from the virtual inertia $Pinertia(t)$, which helps to dampen the frequency oscillations by counteracting the rate of change. This visualization shows the droop control appliances with power variations managing voltage and frequency. It illustrates the grid virtual inertia for enhancing grid stability by mitigating frequency oscillations.

## 4.4. Power-sharing mechanisms

The proposed distributed generation system is composed of several inverter-interfaced sources. Hence, the control strategies must be advanced enough to dynamically distribute power amongst all generation units for each PV output.

**4.4.1. Droop control-based power sharing.** A droop control method is used for low-voltage microgrid or distributed generation systems to share power between inverters. Individually, inverter controls active and reactive power output with local measurements of voltage and frequency:

$$Pshare(t) = Pref - Kp(\omega local(t) - \omega ref) \tag{19}$$

Equation (19) shows that the active power $Pshare(t)$ is adjusted based on the difference between the local frequency $\omega local(t)$ and the reference frequency $\omega ref$. The coefficient $K$ p determines how much the active power output changes in response to frequency deviations.

$$Qshare(t) = Qref - Kq(Vlocal(t) - Vref) \tag{20}$$

Equation (20) shows that reactive power $Qshare(t)$ is adjusted based on the difference between the local voltage $Vlocal(t)$ and the reference voltage $Vref$. The coefficient $Kq$ controls the sensitivity of the reactive power output to voltage changes. Equations (19) and (20) represent active and reactive power sharing among inverters in a microgrid with droop control.

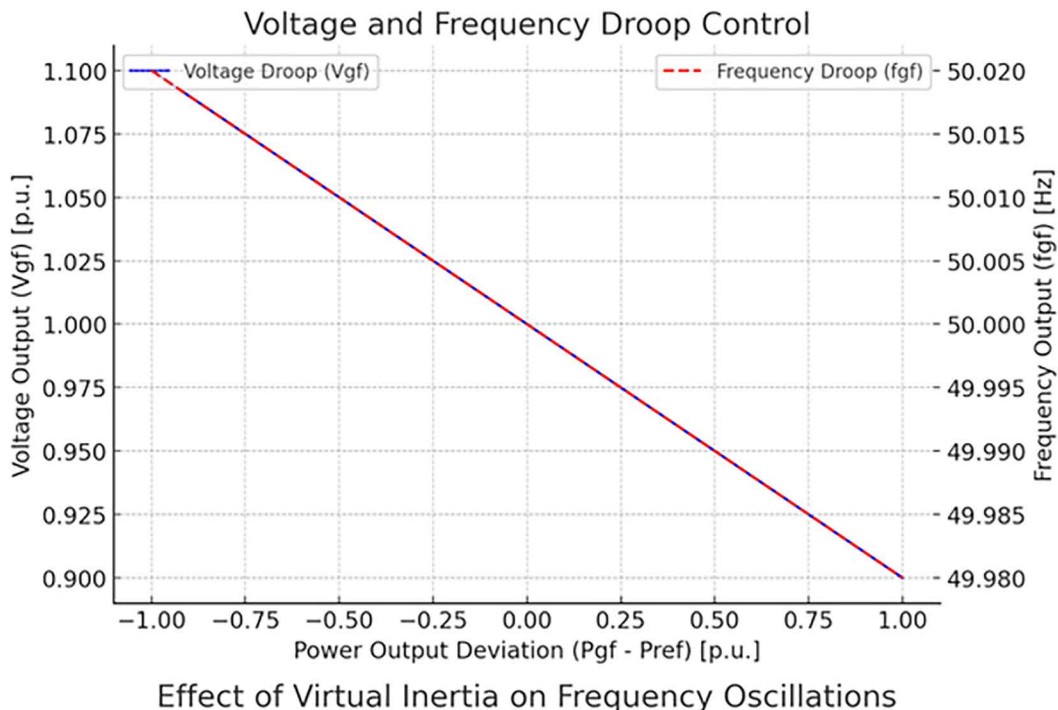

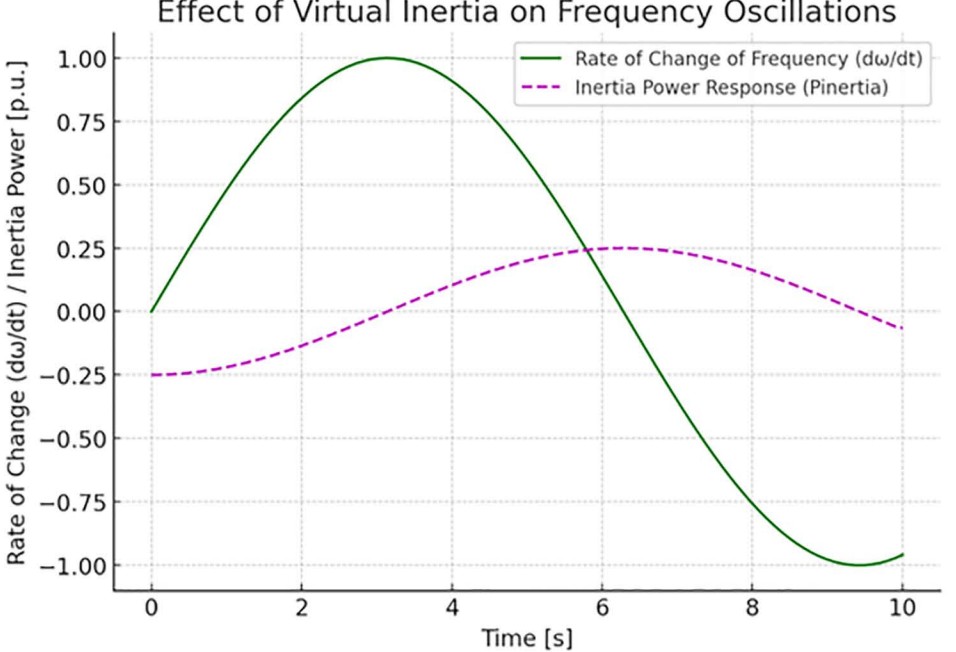

**Fig 5. The voltage and frequency regulation in islanded mode.**

This method operates at the inverter level, where all inverters are leveraged as a separate entity and actively adjust with the degree of generation, along with balancing power sharing without needing any centralized controller.

Fig 6 shows the power-sharing mechanisms based on droop control in a distributed generation system with multiple inverters. The top plot shows the solid blue line representing the active power $P_{share}$ (t) shared by Inverter 1 based on its

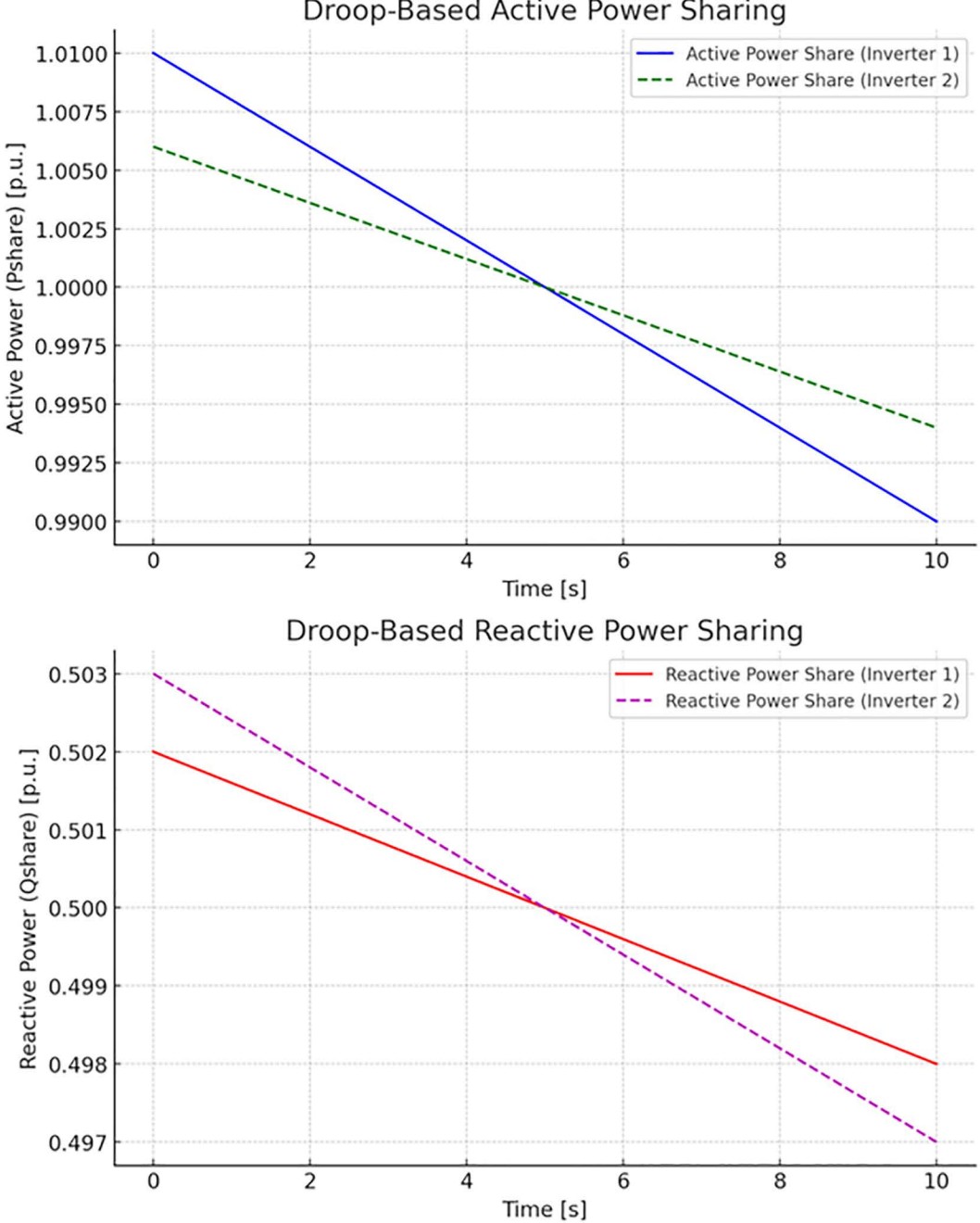

**Fig 6. The power-sharing mechanisms based on droop control in a distributed generation system with multiple inverters.**

local frequency measurements. The dashed green line represents the active power Pshare (t) shared by Inverter 2, which adjusts based on its local frequency. The bottom plot shows the solid red line representing the reactive power Qshare (t) shared by Inverter 1 based on its local voltage measurements. The dashed magenta line shows the reactive power *Qshare* (t) shared by Inverter 2, adjusting based on its local voltage. This visualization illustrates how a microgrid can utilize droop control mechanisms to distribute active and reactive power provided by inverters efficiently and coordinately.

## 4.5. Objective function for sensitivity analysis

The sensitivity analysis affects the system performance of cost, renewable utilization, stability and battery aging of system performance. The analysis uses a baseline case and two supplemental methods, namely One-At-a-Time sweeps of key weights and parameters to isolate immediate influences, along with Monte Carlo-style sampling on a smaller scale for parameter interactions (battery size, communication delay, PV fluctuations, weighting coefficients). The total cost objective J (and \$/day for comparability), the PV utilization (% of available PV energy used vs curtailed), the peak grid imports (kW), the average battery State of Charge (SOC) in percentage, frequency (Hz), number of voltage violations over the horizon and an estimated lifetime for the battery in years from a simple throughput degradation model:

$$\min_x J = \sum_{t=1}^{T} \left[ w_1 C_{loss,t} + w_2 C_{storage,t} + w_3 C_{volt,t} + w_4 C_{freq,t} + w_5 C_{deg,t} + w_6 C_{econ,t} + w_7 C_{comm,t} \right] \tag{21}$$

Equation (21) describes the sensitivity objective function is formulated as a weighted time aggregated cost minimization problem over scheduling horizon T, where control decision variables in x involves inverter setpoints ($P_{i,t}$, $Q_{i,t}$), battery charging and discharging power $P_{b,t}$ and PV curtailment $P_{tcurt}$ and weights $W_k$ sensitivities that are systematically modified to assess the impact on system performance. The cost function consists of several terms: $C_{loss}$,t, the quadratic network line losses weighted by $\alpha\ell$; $C_{storage}$,t capturing charging and discharging costs with cchg and cdis; $C_{volt}$,t penalizing deviation from reference voltage values with $\beta v$; $C_{freq}$,t penalizing deviation from nominal frequency reference by $\beta f$; $C_{deg}$,t the cost of battery degradation based on proportional or rainflow-cycle models; Cecon,t which captures net economic impact of energy market participation and curtailment penalties involving energy prices $\pi tbuy$, $\pi tsell$ and ccurt; and $C_{comm}$, the measure of command tracking errors paid in communication through $\beta comm$. By minimizing these weighted costs together, the sensitivity analysis helps in understanding how different operational objectives, such as loss minimization, economic efficiency, reliability and degradation management, affect the overall system performance. This compact formulation can be directly applied in the linearized power flow approximation, and the problem can be formulated as an explicit quadratic Program with full AC fidelity. The variation of the sensitivity weights and measuring physical parameters is the sensitivity analysis.

### Constraints

$$\text{power balance}: \sum_i P_{i,t} + P_{b,t} - P_{L,t} - P_t^{imp} + P_t^{exp} = 0 \tag{22}$$

$$\text{SOC dynamics}: SOC_{t+1} = SOC_t - \frac{P_{b,t}\Delta_t}{E_{cap}}, SOC^{min} \leq SOC_t \leq SOC^{max} \tag{23}$$

$$\text{inverter limits}: |P_{i,t}| \leq P_i^{max}, |Q_{i,t}| \leq Q_i^{max} \tag{24}$$

$$\text{security limits}: V_n^{min} \leq V_{n,t} \leq V_n^{max}, f^{min} \leq f_t \leq f^{max} \tag{25}$$

Equations (22)-(25) demonstrate the fundamental operational constraints that the optimization framework needs to satisfy. The power balance constraint ensures the equilibrium of the system and requires that the sum of inverter outputs, battery exchange, and imports and exports match the load demand at each time step. The dynamics of the state of charge (SOC) describe the temporal evolution of the battery storage and prescribe that its value at time(t + 1) depends on the previous state of charge subtracted by the normalized charging and discharging power, and remains within SOC limits to ensure

operational safety. The inverter capacity limits both active and reactive power injections to rated values and ensures that no unit operates beyond its physical design. The network security limits ensure that voltage magnitudes and system frequency stays in some allowable bands, and power quality remains stable. The mode logic or droop relations for grid-forming or supporting inverters ensure that the frequency and voltage references remain strictly coupled to power outputs, ensuring system-level stability and flexible operation. The constraints established in these equations secure the technical feasibility, safety, and regulatory compliance of the optimization problem.

## 5. Proposed model

The block diagram of the system and control architecture to optimize power management in a PV-dependent smart grid with grid support and grid-forming inverters is shown in Fig 7. According to the system diagram, the PV generation apparatuses are interfaced by a DC–DC converter and connected to grid-support and grid-forming inverters. The energy storage system is incorporated to smooth out the variability and retain a high-quality supply to the load. The control diagram supports this by illustrating coordination between the optimization algorithm and real-time power management algorithm in control of inverters to facilitate smooth power flow, voltage, and frequency management and to allow a smooth transition between grid-connected and islanded modes. The figures collectively emphasize the interaction of hardware and control in delivering a balance of stability, resilience, and economic operation of distributed renewable energy resources.

## 6. Simulation and case study analysis

The simulation setup and wide-ranging studies from two cases are proposed in this section to confirm the efficiency of those control strategies for PV-based smart grids. The simulations analyze the control, stability, and system response of grid-support and grid-forming inverters, emphasizing power management in grid-connected and island-based modes for multiple operation conditions.

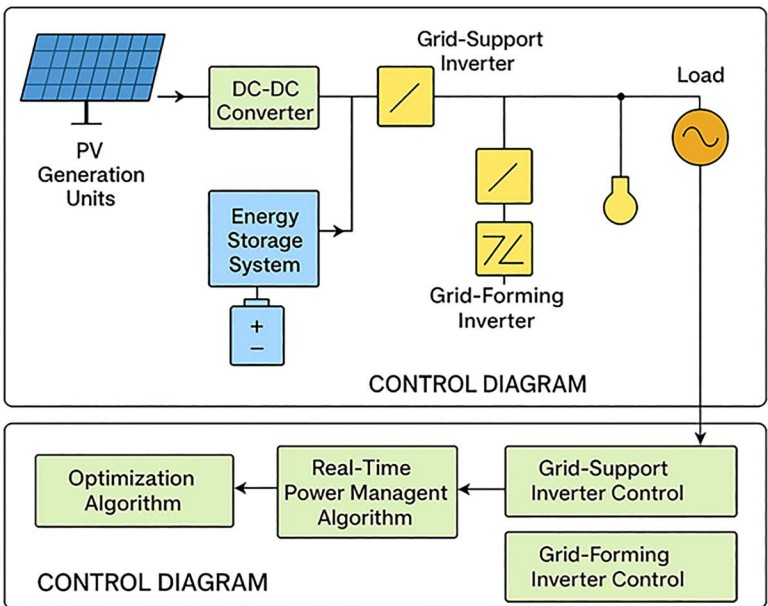

**Fig 7. PV-Based Smart Grid System with grid support and grid-forming inverters.**

## 6.1. Simulation environment

The simulation platform of this study was coded in MATLAB/Simulink, which involved detailed models at the component-level for the PV modules, energy storage system and inverters to assess the proposed power control strategies (Fig 8). The PV modules were represented in terms of a single diode equivalent circuit with irradiance and temperature dependences, for accurate simulation of the actual solar output. The battery energy storage system was signified with the dynamic SOC tracking, round-trip efficiency and cycling degradation models of the battery energy storage system were considered to quantify the availability and lifetime. The inverter models were implemented for grid-support and grid-forming modes with frequency and voltage control via droop characteristics, respectively, allowing analysis of both grid-connected and islanded operation states. The study currently lacks validation against actual hardware experimental data, while these models capture essential behaviours. This introduces some credibility gaps, since practical uncertainties like inverter nonlinearities, communication delay and measurement errors may not be well represented in the simulations. Hardware-in-the-loop testing, a lab-size microgrid demonstrator and other verification with field data from existing PV-based microgrids should be included in future research to increase the reliability and representativeness of the results. The experimental validation of the proposed control strategies would not only show the correctness, but also provide a deeper understanding of the system performance under practical constraints, securing a robustness.

This table ensures reproducibility by providing precise values for system components, enabling other researchers to replicate the study or use similar system setups for comparative analysis.

Fig 8 presents the schematic diagram of a PV-based smart grid simulation environment developed using Simulink tools. It depicts the key components and relations. These allow the recording of grid-feeding power based on irradiance data. The battery dynamically manages its state of charge, charging, or discharging based on grid conditions. Grid-forming/ Grid-support inverter acts as an interface between distributed generation (PV and BESS) and the loads by controlling voltage, frequency, and neutral power flow. Loads signify the power supply of a system based on residential, commercial, or industrial demand. The control algorithm implements real-time power flow management, regulating the inverter and ensuring grid stability. The simulation is executed in Python and other capable software for dynamic smart grid operations. The arrows represent the power flow, and control signals are exchanged between innovative components to show how energy is manipulated and supervised by employing various control strategies. In addition, the system diagram has a parameter data Table (1) that specifies the key values utilized in the study and gives sufficient data for replicating the simulations. The values for the parameters used in Table 1 originated from industry-accepted specifications as well as both established

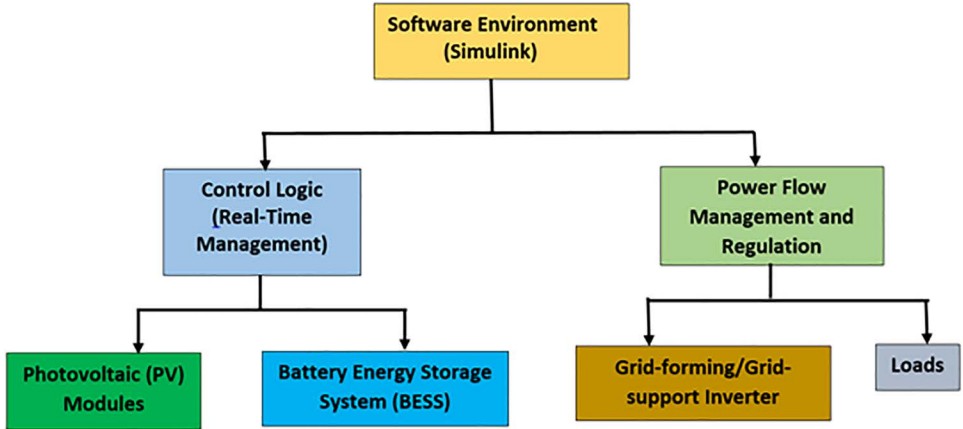

**Fig 8. Simulation schematic model.**

**Table 1. Parameter.**

| Parameter | Value/Description |
|---|---|
| PV Module Power Output | Varies with irradiance (5 kW under peak conditions) |
| Inverter Power Rating (Grid-Support) | 10 kW (maximum active power) |
| Inverter Power Rating (Grid-Forming) | 10 kW (maximum active power) |
| Energy Storage Capacity (BESS) | 20 kWh (total storage capacity) |
| Droop Control Coefficients (Kp) | 0.1 (for frequency regulation) |
| Droop Control Coefficients (Kq) | 0.2 (for voltage regulation) |
| Battery Charge/Discharge Efficiency | 90% (charge), 85% (discharge) |
| Nominal System Voltage | 230 V (AC) |
| Voltage Limits (Vmin, Vmax) | 210 V – 250 V |
| Frequency Limits (fmin, fmax) | 48.5 Hz – 51 Hz |
| Load Demand (Residential) | 5 kW (constant load) |

literature and rational assumptions for purposes of research simulation. For PV module and inverter capacities coincide with those found in residential-scale installations, and the droop control parameters system security limits are based on typical values presented in previous grid-integration works. Battery efficiency values are nominal, and an average residential user represents practical round-trip performance and load profile. These choices ensure a technical realism and analytical tractability for analyzing the optimization framework.

## 6.2. Case study 1: Grid-connected mode

In this scenario, the smart grid operates in grid-connected mode, where distributed PV generation and storage are integrated with the primary grid. The goal is to evaluate the performance of the proposed control strategies under regular operation and grid disturbances.

**6.2.1. Power sharing in grid-connected mode.** The grid--support inverters balance the power between the PV system, energy storage, and the grid. The system uses it to meet local loads, and any surplus is exported back to the grid during peak solar generation. The battery stores excess energy during periods of high PV output and periods of low generation with high demand, and then it is discharged. The power management control algorithms adjust the power flows to distribute generation and demand equally. The simulation shows that the system effectively balances power generation and consumption while energy storage absorbs variations in PV output.

Fig 9 shows a dynamic 24-hour power flow between the PV system, energy storage, and grid in the smart grid environment. The PV power curve (yellow dashed line) begins near 30kW in the morning, increasing throughout the day as solar radiation increases to its maximum amplitude of nearly 45 kW, where it remains constant. This corresponds to the standard daily solar generation profile. The storage power curve (orange dash-dotted line) is initialized at 17 kW, which gradually falls to 10 kW for the remainder of the day, thereby indicating a grid and PV charging pattern. The grid power curve (– red point line) initially is a little below 0 (2 kW), falls further down to nearly −6 kW mid-day and then moves up slightly again to the vicinity of −3 kW by the end of the cycle. It is assumed that the negative values here indicate export to the grid when PV and storage generation exceed local demand. Overall, this figure illustrates how PV generation is dominant in peak irradiance hours. In contrast, storage discharges at a steady pace and the grid is used as a load sink and source when surplus power is being fed into the system or extra power supply is required.

**6.2.2. Disturbance response.** Two types of disturbances are introduced to assess the robustness of the control strategies. Voltage sag is a 10% voltage drop applied to the grid for a short duration. The grid-support inverter detects the sag and injects reactive power to support voltage restoration. Frequency variation for load increasing induces a frequency

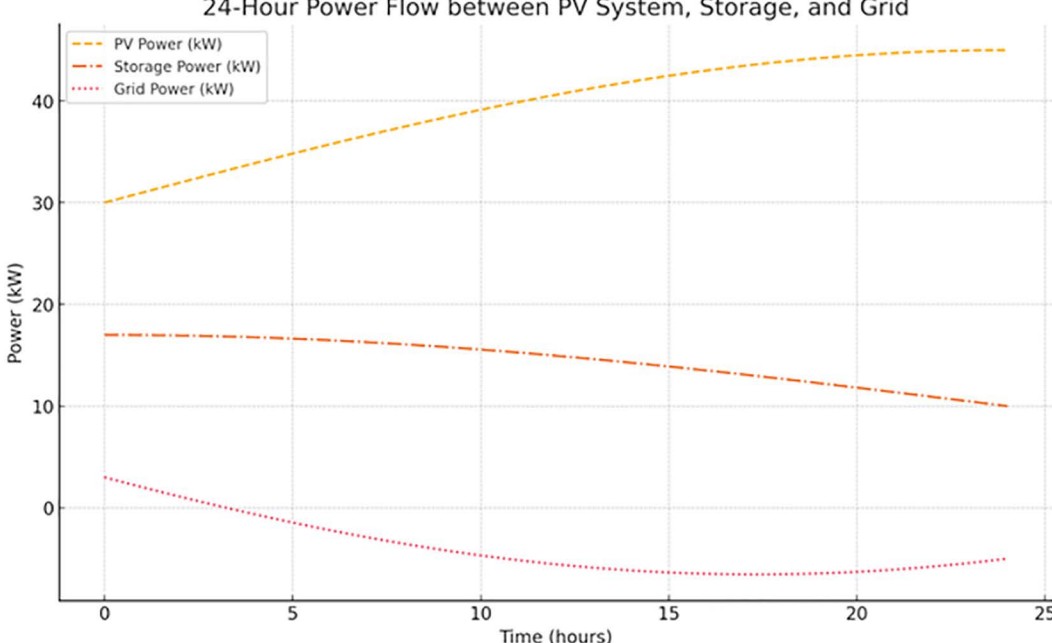

**Fig 9. Power flow dynamics over 24 hours for a grid-connected system.**

drop. The inverter adjusts its active power output based on frequency droop control. The results show that frequency is quickly stabilized within a few seconds as the inverter compensates for the disturbance.

Fig 10 shows that the voltage is restored within a few milliseconds, demonstrating adequate reactive power compensation by the inverter. The inverter adjusts its active power output based on frequency droop control. The results show that frequency is quickly stabilized (within a few seconds) as the inverter compensates for the disturbance. The figure illustrates the system's response to two types of disturbances, such as voltage sag and frequency variation. A 10% voltage drop occurs between 20 ms and 60 ms, simulating a voltage sag. The inverter injects reactive power to help restore the voltage, and the system recovers the voltage within a few milliseconds after the disturbance ends, demonstrating adequate voltage support. A sudden load increase at the 2-second mark causes a frequency drop of 0.2%. The frequency is stabilized quickly (within a few seconds) as the inverter adjusts its active power output using droop control. These responses show the robustness of the control strategies in handling disturbances and maintaining grid stability.

### 6.3. Case study 2: Islanded mode

In the islanded mode, the smart grid operates autonomously without connection to the primary grid. The grid-forming inverters are critical in maintaining voltage and frequency stability in this isolated network.

**6.3.1. Islanded operation under different load conditions.** Different load conditions are exploited to test the control strategies of grid-forming inverters. The battery stores surplus power from PV, and the inverter operates under voltage and frequency to maintain system stability, ensuring low voltage and frequency for light loads. The battery discharges to supply additional power while the inverter adjusts its output to match the heavy load as demand increases.

Voltage and frequency stability under different load conditions in the smart grid are shown in Fig 11. The voltage profile (blue line) of the 10 seconds is overlaid on the top subplot. Voltage varies in the range of around 228 V to 232.5V, and

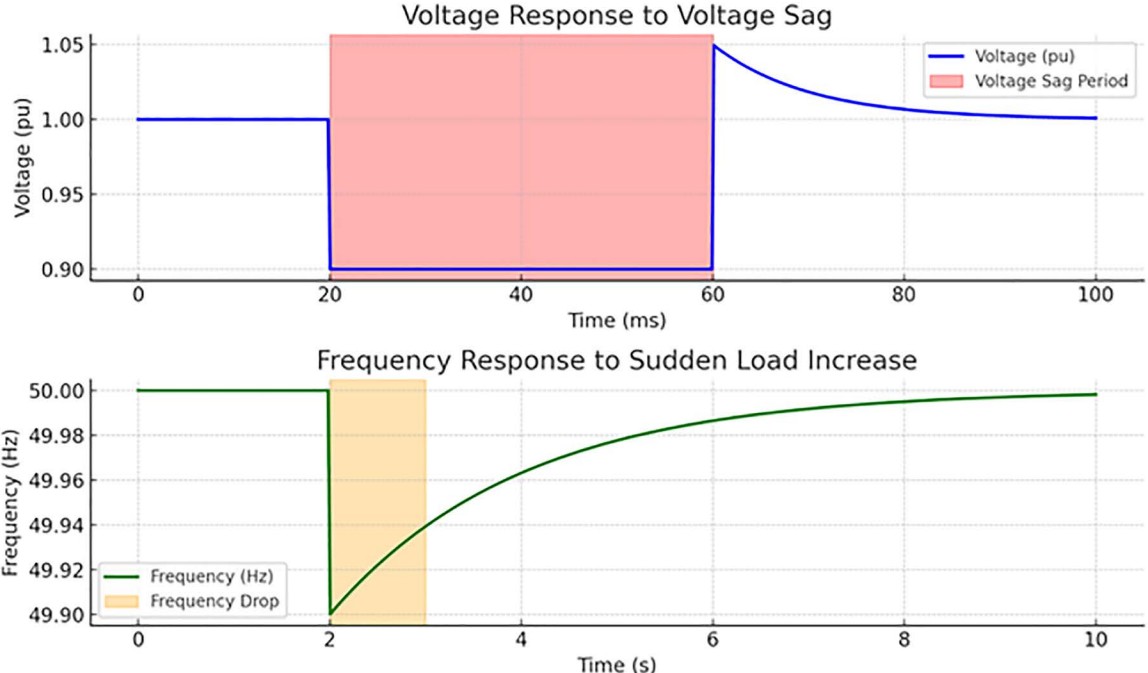

**Fig 10. The grid-supporting inverters maintain stable voltage and frequency with load conditions in grid-connected mode.**

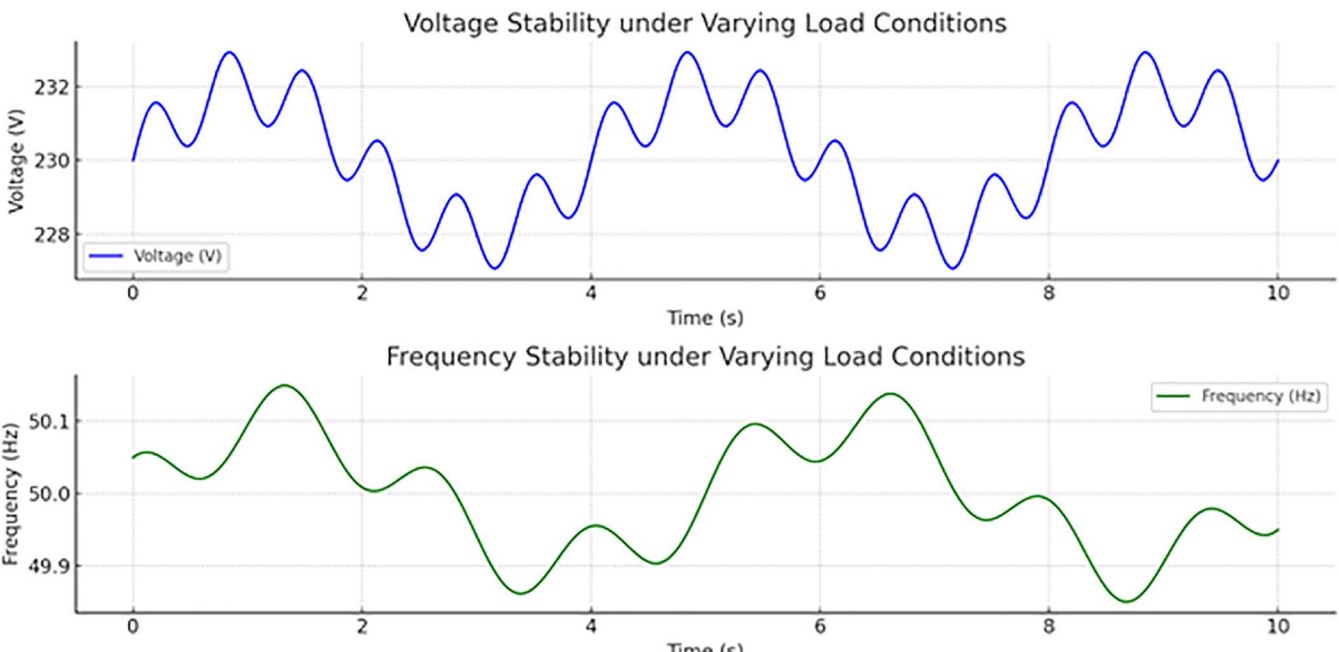

**Fig 11. The grid-forming inverters maintain stable voltage and frequency with load conditions varying in Island mode.**

oscillations arise from the dynamic behavior of the system in response to fast load changes. Nevertheless, the voltage stays in an acceptable operational range even with these disturbances, which proves efficient voltage control mechanisms. The lower subplot shows the frequency response (green line) within the same 10-second window. Frequency oscillates between approximately 49.8 Hz and 50.1 Hz, so minor deviations from the nominal frequency of 50 Hz can be seen. Such disturbances can be anticipated due to the mismatch between generation and demand, but the system is frequency stable without massive error, indicating that the control approach successfully dampens transient oscillations. The overall figure shows that both voltage and frequency stability can be sustained by the system under variable loads, with deviations being well inside normal operating limits—an important feature to prove a reliable coordination of inverters and an effective control logic.

**6.3.2.  Impact of PV output fluctuations.**  Fluctuations in PV generation due to changes in solar irradiance are simulated. The unexpected drop in irradiance, resulting in a 50% reduction in PV output, is applied to simulate cloudy conditions. The battery compensates for the drop by discharging, while the inverter maintains grid stability by adjusting its power output. The PV output rapidly increases, and the system charges the battery to prevent overvoltage. The grid-forming inverter reduces its active power output, preventing system overloading.

Figs 12 and 13 show the battery's state of charge and the inverter's response during the fluctuation, highlighting the system's ability to maintain a stable supply despite changes in PV output. Fig 12 depicts the dynamic behavior of a battery and inverter with PV residuals under fluctuating load, showing that the state of charge (SOC) is oscillating between 70–90% and the output of the inverter is fluctuating between 5KW–35KW over an interval time of 10 seconds, which assures the system can simultaneously balance storage against transformation. Fig 13 two subplots, representing the system stability under a sudden irradiance increasing such as on the top subplot shows voltage jumping from 230V to 234.5V around 2 seconds and then smoothly decaying back to near 230V; the bottom subplot shows a similar transient in frequency, passing slightly up to 50.05 Hz before reaching a settled value of about 50 Hz. These findings demonstrate how energy storage, inverter response time and system stability mechanisms act to dampen sudden variations of PV generation. This shows that the system can effectively achieve voltage and frequency stability under load variations when PV output fluctuates.

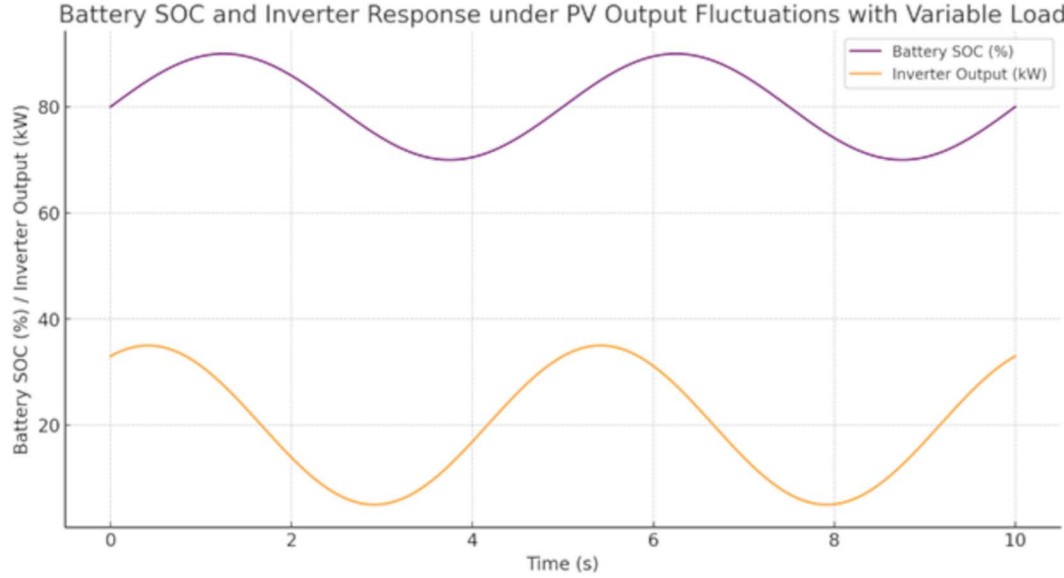

**Fig 12.  Battery and Grid-forming inverter response during the fluctuation in PV output.**

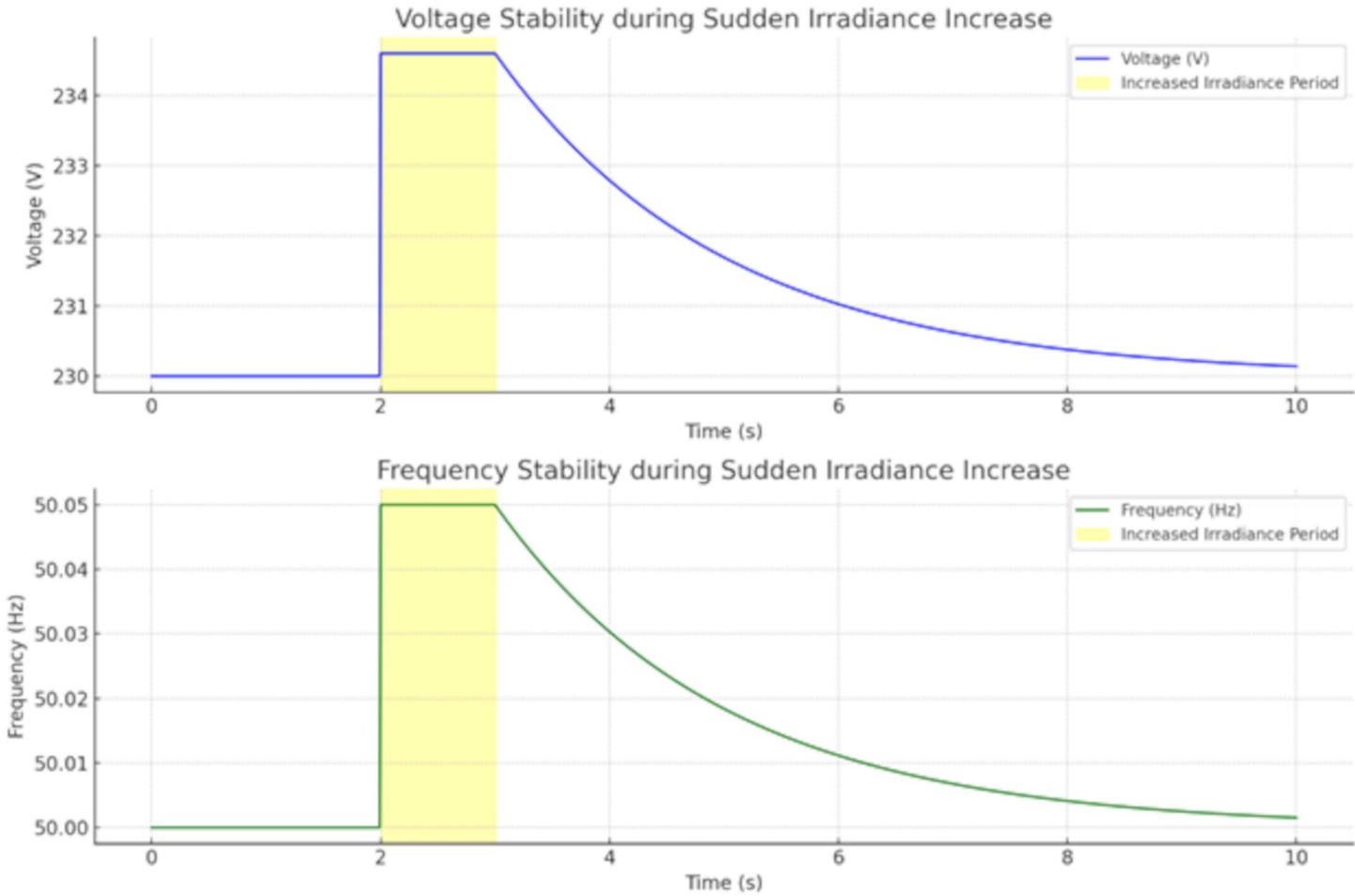

**Fig 13. The inverter maintains voltage and frequency stability during fluctuations in PV output.**

## 6.4. Stress scenarios and robustness tests

This analysis takes into account more realistic stress situations, including extended dips and high step load transients, to verify the robustness of the proposed control loops in harsher operation modes. By evaluating various predisposing performance indices like frequency-based depth, recovery time, state-of-charge margins and over-excessive power import and renewable curtailment levels, it is demonstrated that the billing scalable control action smoothly regulates production schedules to maintain a higher level of operational stability, ensuring reliable power supply and islanding capabilities even at severe disturbances.

**6.4.1. Prolonged low-irradiance.** The irradiance stress profile is obtained by scaling the solar baseline to 15%, 25% and 40% of the peak values with fast cloud dips from more or less clouds with durations of several minutes at levels between 10–30%. The goal of this case is to test storage capacity and verify how the system will behave under long periods of low renewable generation.

Fig 14 shows prolonged low irradiance profiles are presented for a full 48 h of rain and followed by random short cloud dips that test the limits on storage and system resilience. The baseline of irradiances under clear sky (gray dashed) is to show the ideal solar distribution, which peaks at normalized 1. The low-irradiance conditions are normalized to 15%

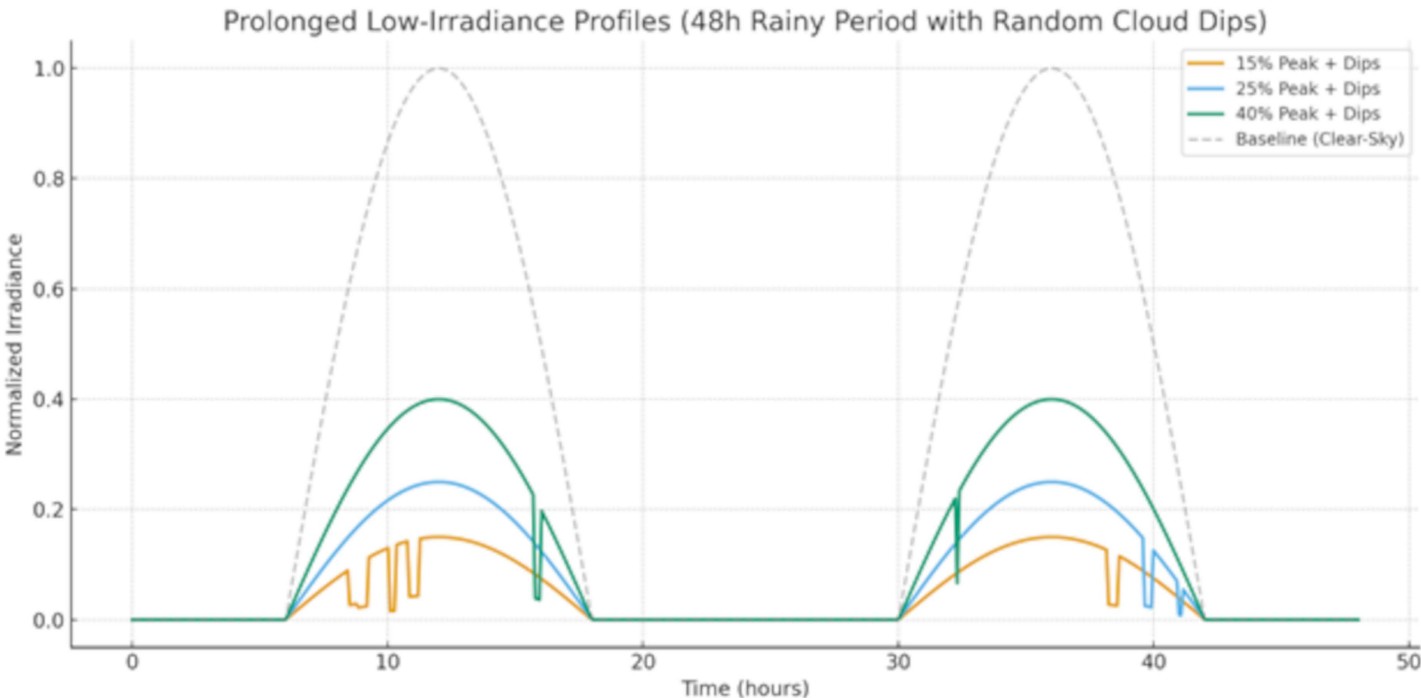

**Fig 14. Prolonged Low-Irradiance Stress Scenario.**

(orange), 25% (blue) and 40% peak of irradiance, representing several days when it is cloudy and raining. Averaged around these lower levels, there are transient short-duration decreases lasting 5–30 min, and attenuating 10–30% from the preceding level; they mimic sudden changes of cloud. This profile reveals a requirement for stable power production period toward longer low-generation periods. That suppression is close to critical for the 15% case that would need a lot of storage and/or grid backup to keep that at equilibrium with demand, and the 40% is definitely better than 30%, but still pretty damn low for dips. The cloud transients superimpose the rapid fluctuations and react against the response of control coordination and inverter coordination. The figure also demonstrates how detrimental sustained irradiance scarcity and random events, which create storage depletion risk, raise reliance on grid import and stability issues – showing that solid energy management must be undertaken alongside appropriate storage capacity dimensioning to achieve resilient microgrid performance.

**6.4.2. Large-scale load variation (demand shock).** The demand shock scenario introduces sudden load step increases of +10%, +30%, +50%, and +75% of baseline demand that last for a duration of 15 minutes, 1 hour and 4 hours are randomly introduced in the load. System resilience is assessed based on frequency nadir, recovery time, peak grid import and islanding capability under extreme demand instabilities.

Fig 15 simulates the effect of demand spikes on a 5 kW load profile during 48 hours. The baseline reflects the overall smooth pattern of a sinusoidal curve, as this can be considered to represent regular daily variations. The discrete load increases of +10, +30, +50 and +75% over baseline are superimposed for 15 min, 60 min or 240 min. Short (15 min) shocks entail sharp but short-lived spikes that medium and long duration (60 min and 240 min) shocks impose increased loads on the system elements for an extensive amount of time, exacerbating system strains. The highest level, condition α +75% 240 min, loads the load nearly 9 kW, which is almost a factor of two higher than for the base and thereby practically possible worst-case test. As this figure shows, system performance depends on the size and duration of demand

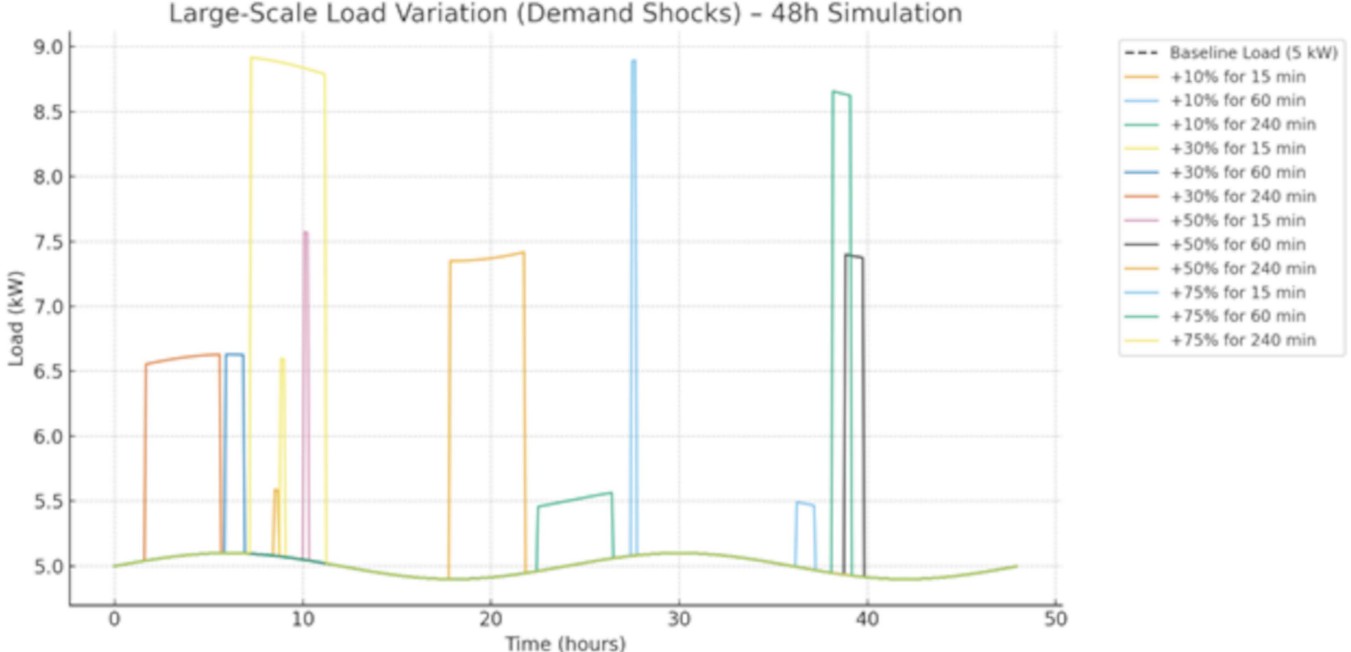

**Fig 15. Large-Scale Load Variation (Demand Shock) Stress Scenario.**

shocks: small, short-term ones can be absorbed at low costs, while larger, more persistent ones may require adequate levels of generation capacity, energy storage or grid stability reserves to absorb them. This simulation highlights the importance of demand response programs and resiliency planning in power systems with rapid load excursions.

## 7. Sensitivity analysis

The sensitivity analysis has been carried out to observe the validity and adaptability of this proposed power management method for different parameters and operating scenarios. The compared sensitivity analysis results on different scenarios illustrate that the variations of frequency weight, degradation weight, battery capacity, communication latency and economic weight impact cost, PV utilization, peak imports, SOC and frequency for battery and battery lifetime.

Fig 16 shows the impact that changes in optimization weights, system size, and communication latency have on system performance from a range of key metrics – cost, PV utilization, peak imports state of charge (SOC), frequency stability, voltage reliability and battery lifetime. The base-case exhibited balanced performance with a daily operational cost of $1,200, PV utilization of 85%, frequency deviation and no voltage violations. When the frequency stability weight was increased, the system's robustness intensified, as the lowest frequency read 49.9 Hz; however, this condition caused an increase in cost ($1,300/day) and a decrease in SOC and battery lifetime. On the other hand, focusing on long-term damage avoidance increased battery life to 12 years and SOC of up to 60% while leading to lower PV utilization and only marginal cost increments. A substantial decrease in battery size resulted in expedited performance degradation, higher costs ($1,500/day), a sharp fall in PV utilization (70%), more voltage violations and shortened life of the storage (6 years), indicating the critical importance of proper sizing. The communication latency was observed to develop, costs increased, and peak imports were limited as a result of contingencies that led to voltage violations -emphasizing the need for real-time communications for inverter coordination. The objective of economic cost minimization was dominated; operating costs were minimized to $1,050/day and provided acceptable performance in almost all the indices, although with little or no reserve when stressed. This finding was validated by the Monte Carlo combined scenario among 200 homogeneously

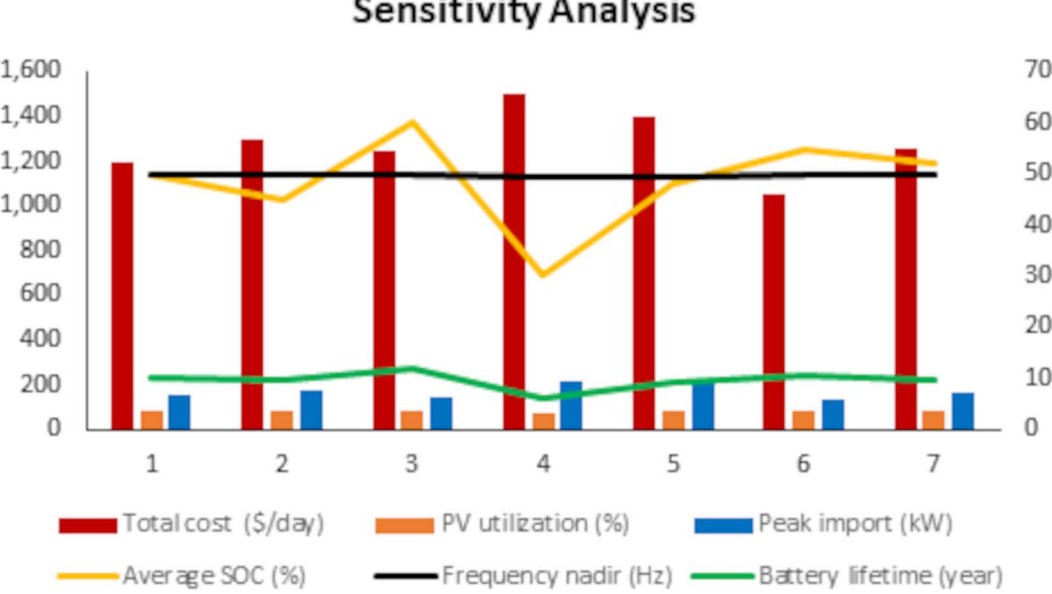

**Fig 16. Sensitivity Analysis.**

distributed random trials, where system median performance comes with modest costs ($1,260/day), 81% PV usage and steady state condition. In general, the sensitivity analysis proves that there exist trade-offs between economic efficiency and resilience and reliability of the PV-based smart grids in terms of economic weightings and degradation weights, although some target benefits with reference to the priorities mentioned earlier can be identified. It is a careful cooperation between economic and degradation weights that need to be delicately balanced to avoid risking undermining system stability and overall reliability.

## 8. Discussion

The proposed control strategies are expected to considerably affect the functioning of high-penetration distributed generation (DG) connected smart grids, especially in the presence of large PV installations. Several possible effects can be identified. Smart grids are conceived to manage the generation, storage and consumption of electricity at home and in neighborhoods like never before, keeping these processes in real-time balance. This conversion to a sharp decrease in the energy loss and less dependence on non-renewable standby systems. Advanced power-sharing strategies can leverage existing distributed generation sources (Photovoltaics, PV) and reduce energy waste. It inherently lowers the utilities' continued operating costs by leveraging infrastructure investments and staving off exorbitant grid capacity expansions. The ability of grid-support inverters to deliver ancillary services in the grid-connected mode improves overall smart-grid resilience, particularly under disturbances such as voltage sags or frequency deviations. That will increase the grid's stability and enable a more robust response in case of contingency events. Proposed control strategies would allow utilities to manage renewable generation effectively, even in locations with higher solar irradiance, and to integrate a higher share of solar power into the grid without compromising stability. Efficient management of variable PV output and power-sharing enhancement among distributed generation units lays a good foundation for the scalability of renewable energy systems. The reliable grid-forming inverters, engineered for island mode operation, allow microgrids to work independently whenever there are faults on the main grids or even during blackouts. This is especially important in places with sparse or sketchy grid infrastructure. These devices feature enhanced control strategies for voltage and frequency

regulation in islanded mode, enabling these regions to become energy-independent using renewable energy sources. The advanced control strategies discussed in this study enable the development of smart grids that are more reliable and efficient, thereby better supporting a sustainable energy system with a high level of distributed renewable generation. The proposed optimization and control decision-making approaches indicate the potential improvements achieved with respect to power flow distribution, system stability, and battery life extension under realistic operating conditions. But perhaps the most interesting attribute is how all these materials hold up under extreme weather. Especially long rainy periods or cloudy periods can drastically decrease PV generation, prompt energy storage systems or grid power to be more involved. While optimal control algorithms that maximize utilization of the battery are employed, capacity constraints may prevent continuous operation under prolonged low-irradiance conditions. This emphasizes the importance of hybrid renewable resources and demand side response that can further enhance resilience under such severe conditions—the impact of inverter coordination delay on system stability. Real-time communication is necessary if active and reactive power sharing, frequency support, and voltage support need to be synchronized in a multi-inverter system. Time delays in the communication and signal processing may result in a misalignment of control actions, resulting in oscillations and deviations from the desired voltage, or even in instability during fast transients. To counteract these impacts, the use of resilient control schemes like adaptive droop control, distributed consensus algorithms, and predictive controllers can improve latency tolerance. Moreover, enhanced communications schemes based on redundancy, higher data rates, and prioritization of critical control signals may contribute to guarantee the coordination's adequacy in very large smart grids. The findings in general emphasize that the proposed systems are capable of dealing with average operating conditions; attention should still be given to extreme weather adaptation and communication-aware control schemes towards more robust, scalable and resilient PV-based smart grids.

## 9. Limitations and recommendations

### 9.1. Limitations

While the research offers substantial improvements in PV-based smart grid management, several challenges and limitations were encountered during the development and simulation phases.

- One of the significant challenges is managing high levels of PV penetration, where rapid fluctuations in PV output due to environmental factors such as cloud cover can destabilize the grid. Although the proposed control strategies effectively mitigate these fluctuations using battery energy storage systems and adaptive inverter control, the reliance on energy storage introduces constraints: the optimal size, cost, and management of these storage systems present ongoing challenges.

- Ensuring grid stability under extreme conditions, such as sudden significant load variations and severe grid faults, remains a technical challenge. While the developed grid-forming inverters perform well under most conditions, there is a risk that during extreme disturbances, such as grid blackouts or prolonged periods of no sunlight, the system might struggle to maintain stability without additional external support or larger energy storage capacity.

- Implementing the real-time control algorithms imparts an evident computational complexity to the system. System stability and high-speed processors or communication infrastructure are needed to manage multiple inverters and distributed generation units simultaneously. In practical aspects, the latency or communication failures due to real-time data exchange between transducers, including inverters, sensors, and controllers, can deteriorate the performance of the control strategies.

- The smart grid will be built of many distributed generation units scattered over a large geographical region, and the control strategies must scale with it as both size and complexity increase. It is particularly difficult to develop scalable methods for the proposed methods on large smart grids and still have nice performance.

- The communication delays could have also been time-dependent and unequal, and the influence of network traffic could undermine the inverter control response and system stability during rapid transients.

- Coordination issues of multiple inverters are one of the practical challenges, as different distributed inverters can have different control parameters, which causes parasitic problems to reactive power sharing, voltage regulation and frequency support when not properly synchronized. The complications are more aggravated in low-communication environments and bandwidth-constrained networks, and ensuring global stability and fairness among multiple distributed energy resources is challenging.

- Extending the simple grid to large smart grids would carry several additional challenges that are out of scope for this work, such as heterogeneous equipment specifications, diverse load profiles, cyber threats and the requirement of hierarchical and multi-layered control architectures. Furthermore, there are currently no hardware-in-the-loop and field tests that can progress towards validating the proposed decentralized technique under realistic operating conditions.

- A critical limitation is related to the expense and size of energy storage. According to the latest intelligence, recent Li-ion battery costs for grid applications are 120–200 USD/kWh, depending upon chemistry, cycle life and vendor [46]. This is a large part of the total system cost, especially in systems with high backup needs, as it tends to be the case for long periods of low irradiance.

- The capacity sizing technique employed in this work was simplified due to the simulation purposes; however, capacitive advanced optimization studies suggest a multi-objective sizing approach aimed at minimizing capital costs, cycle life degradation and system reliability [47]. Approaches to minimization of the optimal solution that support a cost-effective battery size are adopted, and new methods, including stochastic optimization, chance-constrained programming and scenario-based planning, have been incorporated for considering uncertainty in PV output and load fluctuation.

These challenges include control algorithms optimization, storage technology improvement, an advanced communication protocol to be integrated, an adaptive coordination mechanism for the fleets of inverters and experimental validation at a large-scale to guarantee practical deployment in future smart grid environments. Therefore, although the current work shows proof-of-concept feasibility, future work should consider these cost and size constraints by including realistic techno-economic considerations, such as verifying storage needs with actual weather patterns and load peaks, and investigating alternative technologies such as sodium-ion, flow batteries and a hybrid storage systems for cost-effective and scalable deployment.

### 9.2. Recommendations

Recommendations are proposed to improve the robustness, scalability, and practicality of the PV-based smart grid control system for future researchers.

- Firstly, in high-PV penetration conditions, future work will have to consider hybrid energy storage systems with batteries and super capacitors or thermal storage and adaptive forecasting models to reduce the dependency on storage, while also maintaining frequency stability.

- Secondly, to maintain robustness under extreme conditions, it will be important to integrate secondary energy sources (fuel cells or dispatch able renewables) and develop advanced fault-tolerant inverter control schemes to deal with blackouts and long periods of renewable unavailability.

- Thirdly, due to time-sensitive control requirements, distributed and decentralized control structures should be considered, including edge computing and system responsiveness under communication limitations.

- Finally, to support better scalability, the future model should be tested with a hierarchical control architecture and with standardized communication protocols that promote interoperability with a large variety of devices and systems in disparate, geographically dispersed networks.

Further progress in these trends will be necessary to begin deploying smart electric grid systems that are truly resilient, efficient, and at a scale that reflects the promise of renewables.

## 10. Conclusion

This research successfully developed and verified advanced control schemes for improved operation and coordination of PV-centric distributed generation in smart grids. The study provided evidence through extensive simulation results in grid-connected and islanded modes that dynamic power management schemes, grid-support, and grid-forming inverter use-cases, and optimal power flow strategy lead to better power flow management, increased grid stability, and system resiliency enhancement. These are significant contributions to the field since they allow inverters to provide essential grid services such as voltage and frequency Regulation and islanding in outages. This paper offers practical solutions for the microgrid and utility level, enabling the large-scale integration of distributed energy resources. The control strategies maintain efficient power-sharing between PV generation, storage, and loads in grid-connected and islanded modes, thus ensuring improved power management. The power management algorithm optimizes energy usage and minimizes losses by changing power flows based on real-time conditions. While operating in a grid-connected state, the grid-help inverters are successful problem-solving products that sustain consistent links with main power grids under varied voltage and frequency distortions. In islanded operation, grid-forming inverters keep voltage and frequency stable when the load changes, while PV output fluctuates. The control strategies for grid-support and grid-forming inverters enable them to participate in the provision of ancillary services, such as voltage and frequency regulation. The grid-forming inverters are particularly good at supporting grid stability systems in isolated operations. The droop-based power-sharing mechanism provides a fair and efficient load distribution among the distributed generation units, especially during intermittent PV output. The droop-based power sharing supports avoiding system overloads and resource underutilization. The results from these simulations indicate that the proposed control techniques effectively improve the performance and stability of PV-integrated smart grids. These results provide a strong foundation for future research and development to enhance distributed generation systems and inverter technologies for sustainable energy systems. This study further supports the evolution into more intelligent, dependable, and sustainable energy systems in this time of increasing penetration of renewables.

## Author contributions

**Conceptualization:** Amam Hossain Bagdadee.

**Data curation:** Amam Hossain Bagdadee.

**Formal analysis:** Amam Hossain Bagdadee.

**Investigation:** A.K.M. Muzahidul Islam.

**Methodology:** Amam Hossain Bagdadee.

**Software:** Amam Hossain Bagdadee, Ishtiak Al Mamoon.

**Supervision:** Deshinta Arrova Dewi, LI Zhang.

**Validation:** Amam Hossain Bagdadee, LI Zhang.

**Visualization:** Amam Hossain Bagdadee.

**Writing – original draft:** Amam Hossain Bagdadee, Ishtiak Al Mamoon, Deshinta Arrova Dewi, A.K.M. Muzahidul Islam, LI Zhang.

**Writing – review & editing:** Amam Hossain Bagdadee, Ishtiak Al Mamoon, Deshinta Arrova Dewi, A.K.M. Muzahidul Islam, LI Zhang.

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
