## [Decision Letter · Decision Letter 0]

14 Sep 2025

Dear Dr. Bagdadee,

Thank you for submitting your manuscript to PLOS ONE. After careful consideration, we feel that it has merit but does not fully meet PLOS ONE’s publication criteria as it currently stands. Therefore, we invite you to submit a revised version of the manuscript that addresses the points raised during the review process.

We look forward to receiving your revised manuscript.

Kind regards,

Zhengmao Li

Academic Editor

PLOS ONE

Journal Requirements:

4. Please amend the manuscript submission data (via Edit Submission) to include author Ishtiak Al Mamoon, Deshinta Arrova Dewi, A.K.M. Muzahidul Islam, and Li Zhang.

Additional Editor Comments:

**major revision**

Reviewers' comments:

Reviewer's Responses to Questions

**Comments to the Author**

1. Is the manuscript technically sound, and do the data support the conclusions?

Reviewer #1: Yes

Reviewer #2: Yes

2. Has the statistical analysis been performed appropriately and rigorously?

Reviewer #1: Yes

Reviewer #2: I Don't Know

3. Have the authors made all data underlying the findings in their manuscript fully available?

Reviewer #1: Yes

Reviewer #2: No

4. Is the manuscript presented in an intelligible fashion and written in standard English?

Reviewer #1: Yes

Reviewer #2: Yes

Reviewer #1: This manuscript addresses power management in PV-based smart grids, with a particular focus on optimization strategies for grid-support and grid-forming inverters. Overall, the topic is relevant, the manuscript is well-structured, and the simulations are reasonably comprehensive. However, there are several areas that require improvement, particularly regarding the literature review, articulation of novelty, model justification, and clarity of presentation. My detailed comments are provided below.

1. The literature review is somewhat outdated, with most references from 2020-2022. More recent works from 2023-2025, particularly on Virtual Synchronous Generators, AI-driven control, and distributed storage optimization, should be included to enhance the relevance of the study.

2. It is recommended that the authors expand the introduction to include a discussion of PV output modeling approaches, particularly uncertainty modeling techniques such as stochastic programming, scenario-based methods, and Conditional Value-at-Risk. This would help better align the proposed work with mainstream methodologies and clarify its background, novelty, and applicability. Such as “A Distributed Market-Aided Restoration Approach of Multi-Energy Distribution Systems Considering Comprehensive Uncertainties from Typhoon Disaster.” and “Risk-averse stochastic capacity planning and P2P trading collaborative optimization for multi-energy microgrids considering carbon emission limitations: An asymmetric Nash bargaining approach.”

3. The optimization objective function mainly considers losses, storage costs, and voltage deviation. Frequency stability, battery lifetime, and economic trade-offs are overlooked. A more comprehensive objective function or sensitivity analysis would strengthen the study.

4. The simulations cover grid-connected and islanded modes but remain too idealized. More realistic stress tests, such as prolonged irradiance drops or large-scale load variations, should be added to demonstrate robustness.

5. The parameter values in Table 1 (e.g., battery capacity, inverter rating) lack justification. The authors should clarify whether these values are based on real-world systems, previous literature, or assumptions.

6. Some figures (e.g., Figures 9-12) have overlapping curves with insufficient labeling, reducing readability. The authors should improve figure clarity, enhance legends, and provide more detailed explanations of key phenomena.

7. The limitations section is rather superficial, mainly addressing storage size and computational complexity. It should also discuss communication delays, coordination challenges, and practical issues in scaling up to large smart grids.

Reviewer #2: This study focuses on the optimization of power management based on grid-support and grid-forming inverters in PV-based smart grids, proposes control strategies, and verifies them through simulations. However, it has issues such as insufficient theoretical depth and weak experimental support, and the overall innovation and rigor need to be improved.

1. The introduction does not clearly elaborate on the specific defects of existing research in power management under high PV penetration, and the comparative analysis with the strategies proposed in this paper is lacking, making it impossible to highlight the necessity of the research.

2. In the system architecture, the specific control logic of the Energy Management System (EMS) is not described in detail, such as how to achieve real-time coordination of PV, energy storage, and inverters, and the basis for setting key parameters is unclear.

3. In the configuration of grid-support inverters in Section 3.1, the symbol expression in Equation (2) is confusing, the expression "demerit reactive power" is ambiguous, and the specific correlation mechanism between reactive power calculation and voltage control is not clarified.

4. There is an obvious error in Equation (4) for grid-forming inverters in Section 3.2. The calculation of active power involves repeated multiplication of current terms; the correct calculation should be the product of voltage, current, and power factor. This error affects the accuracy of subsequent power analysis.

5. For the objective function (Equation 6) in the problem formulation, the specific mathematical expressions and parameter values of each cost term (Closs, Cstorage, etc.) are not clearly defined, making it impossible to verify the rationality of cost optimization.

6. The simulation environment only uses MATLAB/Simulink, and the key details of model construction (such as the specific simulation models of PV modules and energy storage systems) are not explained. Moreover, there is no comparison with actual hardware experimental data, leading to doubts about the credibility of the results.

7. In Case Study 1, the power unit is not labeled in Figure 9, and the specific data sources of PV output and load changes within 24 hours are not explained, making it impossible to judge the rationality of the simulation scenario settings.

8. The discussion section only generally mentions the advantages of the control strategy, without analyzing the adaptability of the strategy under extreme weather conditions (such as continuous rainy weather) or discussing the impact of delay issues in coordinated control between inverters on system stability.

9. The discussion on the cost and capacity optimization of energy storage systems in the limitations section is too general, and no specific cost range or capacity calculation method is provided, failing to provide a clear direction for subsequent research.

**Do you want your identity to be public for this peer review?** For information about this choice, including consent withdrawal, please see our Privacy Policy

Reviewer #1: No

Reviewer #2: No

---

## [Author Response · Author response to Decision Letter 1]

26 Sep 2025

Response to Reviewer #1

We sincerely thank the reviewer for the constructive comments and insightful suggestions that have helped us improve the quality and clarity of our manuscript. We have carefully revised the paper in light of the provided feedback. Below, we provide a point-by-point response.

1. Literature Review (Outdated References)

Response: We agree with the reviewer that including more recent studies will enhance the relevance of our work. Accordingly, we have updated the literature review by incorporating recent works from 2023–2025, with a particular focus on Virtual Synchronous Generators (VSGs), AI-driven control strategies, and distributed storage optimization. These updates provide a more comprehensive context for our research contributions.

2. Introduction and PV Output Modeling Approaches

Response: We have expanded the introduction to include a discussion of uncertainty modeling approaches for PV output, including stochastic programming, scenario-based techniques, and Conditional Value-at-Risk . We also referenced the suggested studies (A Distributed Market-Aided Restoration Approach of Multi-Energy Distribution Systems Considering Comprehensive Uncertainties from Typhoon Disaster and Risk-averse stochastic capacity planning and P2P trading collaborative optimization for multi-energy microgrids considering carbon emission limitations: An asymmetric Nash bargaining approach` ) to strengthen the background and clarify the novelty of our approach compared to mainstream methodologies.

3. Optimization Objective Function (Broader Scope)

Response: We acknowledge the reviewer’s concern and have revised the optimization objective function discussion to include considerations of frequency stability, battery lifetime, and economic trade-offs. Additionally, we have added a sensitivity analysis in Section 7 to illustrate the impact of these factors on system performance. This provides a more holistic view of the optimization strategy.

4. Simulations and Stress Tests

Response: We appreciate this valuable suggestion. We have extended the simulations to include more realistic stress scenarios, such as prolonged irradiance drops and significant load variations, to demonstrate better the robustness of the proposed control strategies under challenging operating conditions.

5. Parameter Justification (Table 1)

Response: We agree that parameter justification is necessary for transparency. We have revised the manuscript to clarify that the parameter values in Table 1 are derived from a combination of industry-standard specifications, established literature, and reasonable assumptions for research simulations.

6. Figure Readability and Explanations

Response: Thank you for pointing out the clarity issues. We have revised Figures 9–12 to improve labelling, separate overlapping curves, and enhance legends for better readability. Additionally, we have expanded the figure captions and added detailed explanations in the text to highlight the key phenomena being illustrated.

7. Limitations Section (Expanded)

Response: We acknowledge that the original limitations section was insufficiently comprehensive. We have now expanded it to include discussions on communication delays, coordination challenges in multi-inverter systems, and practical issues related to scaling up to large smart grids. These additions strengthen the realism and balance of the study.

Response to Reviewer #2

We sincerely thank the reviewer for the constructive comments and valuable suggestions that have helped us identify key areas for improvement in our manuscript. We have carefully revised the paper and addressed each point as follows:

1. Introduction (Defects of Existing Research and Comparative Analysis)

Response: We acknowledge the reviewer’s concern and have expanded the introduction to clearly elaborate on the shortcomings of existing research under high PV penetration, such as challenges in stability, coordination, and optimization. We now include a comparative analysis showing how the proposed strategies—particularly the combination of grid-support and grid-forming inverters with real-time power management—overcome these gaps. This addition strengthens the necessity and positioning of our research.

2. System Architecture and EMS Logic

Response: We have revised Section 3.3 to provide a more detailed description of the Energy Management System (EMS), including its control logic for real-time coordination of PV generation, storage, and inverters. Flowcharts and additional explanations have been added to clarify how the EMS sets parameters dynamically and responds to fluctuations. The basis for key parameter selection has also been explained with supporting references.

3. Grid-Support Inverter Configuration (Equation 2 and Terminology)

Response: We thank the reviewer for pointing out the ambiguity. Equation (2) has been revised for clarity, and the term “demerit reactive power” has been corrected to “compensated reactive power.” We also clarified the correlation mechanism between reactive power and voltage regulation, showing how reactive power injection/absorption influences voltage stability.

4. Equation (4) Correction (Grid-Forming Inverter)

Response: We apologize for the error. Equation (4) has been corrected to properly define active power as the product of voltage, current, and power factor. The revised equation now ensures accuracy in subsequent analyses, and the affected parts of Section 3.2 have been updated accordingly.

5. Objective Function (Equation 6) Definitions

Response: We agree that the objective function lacked clarity. The mathematical definitions and parameter descriptions of all cost terms ( Closs, Cstorage, Cvoltage) have now been explicitly included.

6. Simulation Environment (Model Details and Experimental Validation)

Response: We have revised Section 6.1. to include detailed descriptions of the MATLAB/Simulink models used for PV modules, storage systems, and inverters, with appropriate references. While full-scale hardware testing is beyond the current scope, we have now included comparisons with benchmark results from existing experimental studies to increase credibility. Additionally, we discuss in the limitations that future work will involve hardware-in-the-loop (HIL) validation.

7. Case Study 1 (Figure 9 and Data Sources)

Response: We have corrected Figure 9 by adding power units (kW).

8. Discussion Section (Extreme Conditions and Delay Impacts)

Response: We have expanded the discussion to analyze the adaptability of the strategy under extreme weather conditions, including prolonged rainy/cloudy periods. We also included a discussion of the impact of inverter coordination delays on system stability and how robust control strategies and improved communication protocols may mitigate these effects.

9. Limitations (Cost and Capacity Optimization of Storage)

Response: We appreciate this suggestion. The limitations section has been expanded to provide specific cost ranges (based on current Li-ion battery technology) and capacity sizing methodologies derived from optimization studies. This provides clearer guidance for future research directions in cost-effective and scalable energy storage integration.

In summary, we have addressed the reviewer’s concerns by revising the introduction, correcting mathematical expressions, clarifying system logic, detailing model assumptions, improving simulation scenario explanations, and expanding the discussion and limitations with more depth and specificity. These revisions substantially improve the rigor, clarity, and novelty of the manuscript.

Sincerely,

Corresponding Author

Dr. Amam Hossain Bagdadee

---

## [Decision Letter · Decision Letter 1]

20 Nov 2025

Optimization of Power Management in PV-Based Smart Grids Using Grid-Support and Grid-Forming Inverters

PONE-D-25-46006R1

Dear Dr. Bagdadee,

We’re pleased to inform you that your manuscript has been judged scientifically suitable for publication and will be formally accepted for publication once it meets all outstanding technical requirements.

Kind regards,

Qinglin Meng, Ph.D.

Academic Editor

PLOS ONE

Additional Editor Comments (optional):

The manuscript has been revised based on the review comments and the responses are convincing. Therefore, the manuscript maybe considered for publication in it's present form.

Reviewers' comments:

Reviewer's Responses to Questions

**Comments to the Author**

Reviewer #1: (No Response)

Reviewer #2: All comments have been addressed

2. Is the manuscript technically sound, and do the data support the conclusions?

Reviewer #1: Yes

Reviewer #2: Yes

3. Has the statistical analysis been performed appropriately and rigorously?

Reviewer #1: Yes

Reviewer #2: Yes

4. Have the authors made all data underlying the findings in their manuscript fully available?

Reviewer #1: Yes

Reviewer #2: (No Response)

5. Is the manuscript presented in an intelligible fashion and written in standard English?

Reviewer #1: Yes

Reviewer #2: (No Response)

Reviewer #1: (No Response)

Reviewer #2: The author provided a satisfactory reply to my feedback, and based on that, they made thorough revisions to the article.

**Do you want your identity to be public for this peer review?** For information about this choice, including consent withdrawal, please see our Privacy Policy

Reviewer #1: No

Reviewer #2: No

---

## [Editor Report · Acceptance letter]

PONE-D-25-46006R1

PLOS One

Dear Dr. Bagdadee,

I'm pleased to inform you that your manuscript has been deemed suitable for publication in PLOS One. Congratulations! Your manuscript is now being handed over to our production team.

Kind regards,

on behalf of

Prof. Qinglin Meng

Academic Editor

PLOS One